**METHOD**                                                                                              **Open Access**

# TADA—a machine learning tool for functional annotation-based prioritisation of pathogenic CNVs

Jakob Hertzberg[1,2]* , Stefan Mundlos[1,2], Martin Vingron[1] and Giuseppe Gallone[1]

*Correspondence:
hertzber@molgen.mpg.de
[1] Max Planck Institute for Molecular Genetics, Ihnestraße 63, 14195 Berlin, Germany
Full list of author information is available at the end of the article

## Abstract

Few methods have been developed to investigate copy number variants (CNVs) based on their predicted pathogenicity. We introduce TADA, a method to prioritise pathogenic CNVs through assisted manual filtering and automated classification, based on an extensive catalogue of functional annotation supported by rigourous enrichment analysis. We demonstrate that our classifiers are able to accurately predict pathogenic CNVs, outperforming current alternative methods, and produce a well-calibrated pathogenicity score. Our results suggest that functional annotation-based prioritisation of pathogenic CNVs is a promising approach to support clinical diagnostics and to further the understanding of mechanisms controlling the disease impact of larger genomic alterations.

**Keywords:** Copy-number-variants, Structural variants, Pathogenicity prediction, Functional annotation, TADs, Machine learning

## Background

The investigation of the genetic causes of rare developmental disorders and, ultimately, the molecular diagnosis of rare disease patients relies on the accurate detection and prioritisation of disease-causing DNA variants. It follows that the accurate identification and prioritisation of candidate disease-associated genetic variation is a fundamental question in human genetic research. The disease impact of single nucleotide polymorphisms (SNPs) and small Insertions and Deletions (InDels) has been the focus of extensive study [1–3]. Comparatively, little is known about the mechanisms and disease impact of structural variants (SVs), including unbalanced SVs, also known as copy number variants (CNVs). CNVs have significant and pervasive impact on phenotypic variability and disease: they can affect gene dosage [4] and modulate basic mechanisms of gene regulation [5]. In addition, CNVs have been shown to disrupt topologically associating domains

(TADs) [6] and can rewire long-range regulatory architectures, resulting in pathogenic phenotypes [7, 8].

One of the reasons why CNVs are poorly understood is because they are difficult to reliably detect, filter and interpret given current sequencing technology. New experimental approaches such as long-read sequencing [9] combined with novel, long-read specific algorithms for read alignment [10, 11] and SV detection [11–13] are allowing a more thorough survey of the spectrum of large variation in healthy and diseased human genomes [14, 15]. This raises the need for methods to interpret, score and prioritise SVs to support clinical practice.

Ongoing efforts to annotate the potential contribution of SVs to disease suggest the possibility of using functional annotation to stratify SV calls by relevance and/or predicted pathogenic potential [16]. In terms of tools and methods to prioritise pathogenic CNVs, a number of approaches have been proposed. ClinTAD [17] and TADeus [18] focus on providing a visual framework to aid a human expert in manually surveying and flagging likely relevant SVs. The Variant Effect Predictor (VEP) [19] allows for the annotation of SNPs as well as CNVs and returns an approximation of the variant's impact. SVScore [20] aggregates SNP-level CADD [21] scores, integrating single nucleotide-based deleteriousness prediction over the length of a SV. This is based on the assumption that SV effects can be thought of as agglomerates of single-nucleotide-level effects, which is generally unlikely to be the case, given growing evidence of SV impact on, for instance, gene dosage [4] and regulatory context [5]. Kumar et al. recently introduced SVFX [22] , a machine learning framework to quantify pathogenicity for somatic and germline CNVs. SVFX represents, to our knowledge, the first flexible machine learning-based model for SV pathogenicity prediction. In their analysis the authors rely mainly on somatic variants as a proxy for pathogenicity as opposed to a set of germline variants annotated as pathogenic. While a subset of these variants is actually pathogenic, the model still likely trains on the differences between somatic and common germline mutations, rather than pathogenic versus non-pathogenic. The SVFX authors also provide germline models for specific disease contexts and ClinVar variants. However, their models are limited by a major aspect of the modelling procedure: SVFX employs a normalisation method causing information leakage between training and test data and includes SV size as a feature leading to an overestimation of their performance on germline variants—especially small to medium sized SVs.

Here, we present the TAD annotation tool (TADA), a method to annotate CNVs in the context of their functional environment, based on a rich set of coding as well as noncoding genomic annotation data. The annotation data is centred around TAD boundaries, which serve both as proxy for the regulatory environment (in that they limit the genomic annotation potentially affected by the CNV to the loci between boundaries) and as annotation themselves. TADA is designed to assess the functional relevance of user-specified input sets of CNVs of unknown clinical relevance by one of two methods: (a) annotation, followed by manual filtering, or (b) machine learning-based automated classification. Importantly, our machine learning models for duplications and deletions are trained on a set of annotated pathogenic variants (DECIPHER [23]) and rigorously driven by functional evidence: we carefully assess the potentially discriminating effect of each of the annotations considered by performing enrichment tests, comparing the expected and observed overlap of pathogenic versus non-pathogenic CNVs and functional annotation

data. We demonstrate the applicability of our approach on two separate test sets: (1) a set of *ClinVar* variants, a database comprised of curated variants from multiple studies annotated with their respective clinical significance, and (2) a dataset composed of clinically validated pathogenic DECIPHER and curated non-pathogenic variants not included in our training data, resulting in an ROC-AUC for the deletion model of 0.8059 and 0.8865, respectively. Both the deletion and the duplication model correctly rank more than 30% of pathogenic variants at the first 5 of 100 ranks based on predicted pathogenicity when compared to 99 non-pathogenic variants. Our approach outperforms alternative current methods, namely SVScore, SVFX, CADD-SV and VEP, in direct classification measured by AUC and F1-score. TADA is available free-of-charge under the MIT license and can be customised for prioritising or classifying CNVs from different disease contexts.

## Results

### Enrichment analysis of pathogenic CNVs

We performed a comprehensive enrichment analysis of the pathogenic DECIPHER variant data set [23] in comparison to a curated set of common, and therefore unlikely pathogenic, CNV calls [14, 15, 24, 25]. Our purpose was to assess whether we could identify contrasting patterns of enrichment/depletion in a pathogenic set with respect to a control set. We reasoned that, if this was the case, the discriminating annotations would be excellent candidates for a feature set of a classifier to distinguish pathogenic from non-pathogenic variants. In our analysis, we account for size differences between the variant sets and for the non-uniform mutation rate across the genome [15] (which would have artificially inflated fold changes) by building GC-content isochores [26] and constraining bootstrapping with bins of comparable GC-content signal (Methods for details). The set of annotations tested in the enrichment analyses was based on evidence from Collins et al. 2019 [14] and Audano et al. 2019 [15] including coding and non-coding annotation as well as conservation and predicted loss-of-function (pLoF) metrics. We additionally integrated TAD boundaries [27], CTCF bindings sites [28], genes associated with developmental disease (DDG2P) as well as genes predicted to be haplosufficient (HS Genes) and haploinsufficient (HI Genes) [4]. The results for pathogenic and non-pathogenic deletions are shown in Fig. 1.

We conducted our enrichment tests within the genome association tester (GAT) framework [29], a bootstrap-based method to test for enrichment or depletion of genomic segments in background annotations accounting for a variety of confounding factors (Methods for details). Briefly, we generated a number of randomly distributed, size-matched genomic segments in each simulation and computed overlaps with sets of genomic annotation. We computed overlaps over all simulations (*expected* overlaps, see Methods) and compared them to *observed* overlaps, producing fold-change (FC) values and an empirical *p* value showing the associated significance for each annotation set. The *q* value refers to a multiple testing corrected *p* value [30]. Accepting anything below a certain *q* value threshold corresponds to controlling the false discovery rate at that level. To account for potentially diverging patterns of enrichment/depletion due to the variant type, we ran enrichment tests separately for deletions and duplications using 10,000 simulations.

In agreement with what was previously shown by Collins et al. 2019 we observe significant depletion of non-pathogenic deletions in coding regions (logFC = − 0.340, *q*-val =

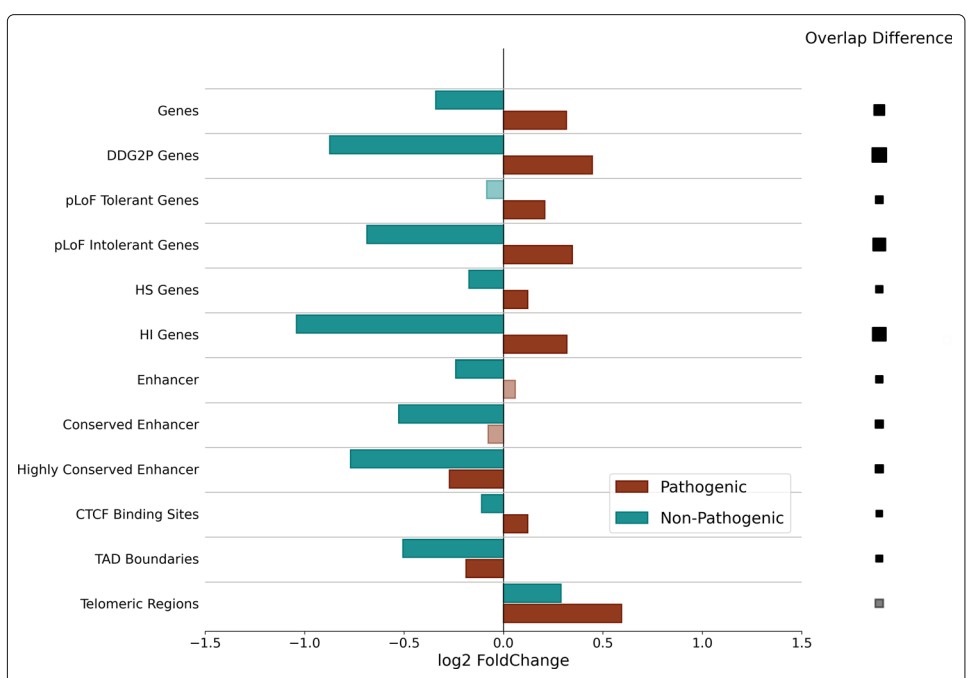

**Fig. 1** Enrichment Analysis of non-pathogenic and pathogenic deletions. The figure shows the $\log_2$(fold change) for expected and observed variant overlap for each set of genomic annotations based on 10,000 simulations. The size of the squares on the right side of the figure is proportional to the overlap FC difference between pathogenic and non-pathogenic deletions. Grey bars and squares indicate a non-significant FC ($q$ value $\leq 0.01$)

$0.15 * 10^{-3}$) and regulatory regions (logFC = $-0.240$, $q$-val = $0.15 * 10^{-3}$). The depletion signal increases with predicted haploinsufficiency (logFC = $-1.041$, $q$-val = $0.15*10^{-3}$) of the affected coding regions and conservation of the regulatory regions (logFC = $-0.768$, $q$-val = $0.15 * 10^{-3}$). While we observe a strong significant depletion of non-pathogenic deletions in pLoF intolerant genes, we do not detect significant depletion in pLoF tolerant genes. We observe stronger enrichment in pLoF intolerant genes with respect to background gene annotation (logFC = $-0.686$, $.0.67 * 10^{-3}$), confirming previous observations reporting increased depletion of common structural deletions in coding regions intolerant to LoF-mutations. In agreement with Audano et al. 2019 [15] we observe significant enrichment of non-pathogenic deletions in extended (see Methods for an definition of *extended*) telomeric regions (logFC = $0.289$, $q$-val = $0.8727 * 10^{-3}$). Additionally, our combined set of non-pathogenic deletions is significantly depleted in TAD boundaries (logFC = $-0.506$, $q$-val = $0.15 * 10^{-3}$) and CTCF binding sites (logFC = $-0.109$, $q$-val = $0.72 * 10^{-3}$).

The enrichment analysis of pathogenic DECIPHER deletions reveals patterns of significant enrichment in all functional annotation except FANTOM5 enhancer regions, TAD boundaries and extended telomeric regions. The pathogenic deletions are significantly enriched in coding regions (logFC = $0.316$, $q$-val = $0.15*10^{-3}$), with increased enrichment for DDG2P genes (logFC = $0.447$, $q$-val = $0.15*10^{-3}$). We observe increased enrichment in pLoF intolerant genes (logFC = $0.346$, $q$-val = $0.8*10^{-3}$) compared to pLof tolerant genes (logFC = $0.208$, $q$-val = $.15 * 10^{-3}$) as well as HI genes (logFC = $0.319$, $q$-val = $0.15 * 10^{-3}$) compared to HS genes (logFC = $0.122$, $q$-val = $0.13 * 10^{-3}$). The pathogenic deletions are

also significantly enriched in extended telomeric regions (logFC = 0.594, $q$-value = 0.15 $*$ $10^{-3}$). We are not able to detect significant enrichment of our pathogenic set in any set of the enhancer annotations, regardless of the extent of sequence conservation. Instead, we observe significant depletion of pathogenic deletions in highly conserved enhancers (logFC = $-$ 0.272, $q$-val = 0.15 $* 10^{-3}$). Even though CNVs have been shown, as previously mentioned, to cause disease phenotypes by disrupting TAD boundaries, the enrichment analysis reveals a significant depletion of pathogenic deletions in TAD boundaries (logFC = $-$ 0.188, $q$-val = 0.15 $* 10^{-3}$). The analysis of duplications reveals similar patterns of enrichment for pathogenic and non-pathogenic variants (Additional file 1: Fig. S1).

Doane et al. 2016 observed an enrichment of CNVs impacting human accelerated regions (HARs) [31] i.e. regions that are highly conserved across vertebrates with increased divergence in humans, in individuals with autism spectrum disorder (ASD). This suggests potential brain associated regulatory function of HARs [31]. To test a wider range of genomic annotation such as HARs as distinguishing features, we set out to conduct further enrichment analyses. Motivated by evidence of CNV enrichment in segmental duplications (SDs) [32], we included SDs in our analysis. Given the increasing evidence for the impact of non-coding variation in Mendelian disorders, localising in highly conserved, tissue-specific active distal regulatory elements such as enhancers [33], we also included ChromHMM annotations [34]. The enrichment results of SDs, HARs and ChromHMM annotation for deletions and duplications are shown in Fig. S2 and S3 (Additional file 1), respectively. Both non-pathogenic and pathogenic deletions are significantly depleted in polycomb-repressed regions (logFC = $-$ 0.477, $q$-val = 0.7$*10^{-3}$, logFC = $-$ 0.168, $q$-val = 0.14$*10^{-2}$, respectively) and HARs (logFC = $-$ 0.399, $q$-val = 0.1$*10^{-3}$, logFC = $-$ 0.216, $q$-val = 0.1 $* 10^{-3}$, respectively). We observe no significant depletion or enrichment of pathogenic or non-pathogenic deletions in any other ChromHMM annotation or in SDs. In contrast we observe significant enrichment of non-pathogenic duplications in SDs (logFC = 0.375, $q$-val = 0.1 $* 10^{-3}$). We also observe significant depletion of non-pathogenic duplications in polycomb-repressed regions (logFC = $-$ 0.248, $q$-val = 0.1 $* 10^{-3}$) and small but significant depletion in HARs (logFC = $-$ 0.560, $q$-val = 0.1 $* 10^{-3}$).

TADs are known to approximately represent windows of constrained regulatory interactions [35]. We reasoned that for TADs of high regulatory relevance, pathogenic CNVs are likely depleted across the entire TAD environment due to their potential effect on the corresponding regulatory context. We therefore set out to investigate the enrichment of pathogenic CNVs in TADs stratified by their regulatory importance. We assumed that the regulatory importance of a TAD can be approximated by the conservation of enhancer annotation and the pLoF intolerance of coding annotation within the TAD environment (Methods for details). We henceforth refer to the resulting set of TAD annotations as *TAD-centric* annotations. Figure S4 and S5 (Additional file 1) show the results of the TAD-centric enrichment analysis for, respectively, deletions and duplications. We observe significant enrichment of non-pathogenic deletions and duplications in TADs lacking known coding or regulatory annotation (logFC = 1.128, $q$-val = 0.47$*10^{-3}$, logFC = 1.436, $q$-val = 0.15 $* 10^{-3}$, respectively) and significant depletion of non-pathogenic CNVs in most TADs containing coding or regulatory elements. In contrast, we observe significant enrichment of pathogenic deletions in TADs containing coding and regulatory annotation with an increased enrichment in TADs encompassing at least one highly loF intolerant

gene (logFC = 0.240, $q$-val = 0.28 * $10^{-3}$). However, we do not detect a signal of enrichment or depletion for pathogenic duplications in TAD-centric annotations and cannot confirm increased enrichment of pathogenic deletions in TADs containing highly conserved enhancers compared to TADs with less conserved enhancers. Taken together, the results point towards selective pressure towards deletions based on the entire affected regulatory domain rather than individual coding and non-coding annotation.

## TADA

We used the observed patterns of enrichment and depletion to inform feature selection in our TAD annotation tool. However, we are well aware that the relevance of the selected features for the prioritisation of putative pathogenic variants will differ, based on disease and sample context. To account for the variable relevance of features we allowed user-defined annotation alongside the default feature set driven by the results of the enrichment analysis. A schematic and a detailed description of the TADA workflow is shown in Fig. 2.

## Pathogenicity Prediction

We demonstrate the viability of functional annotation as a basis for putative pathogenic CNV prioritisation by training classifiers using the TADA tool and evaluating their predictive performance. We split the variant set used for the enrichment analyses into

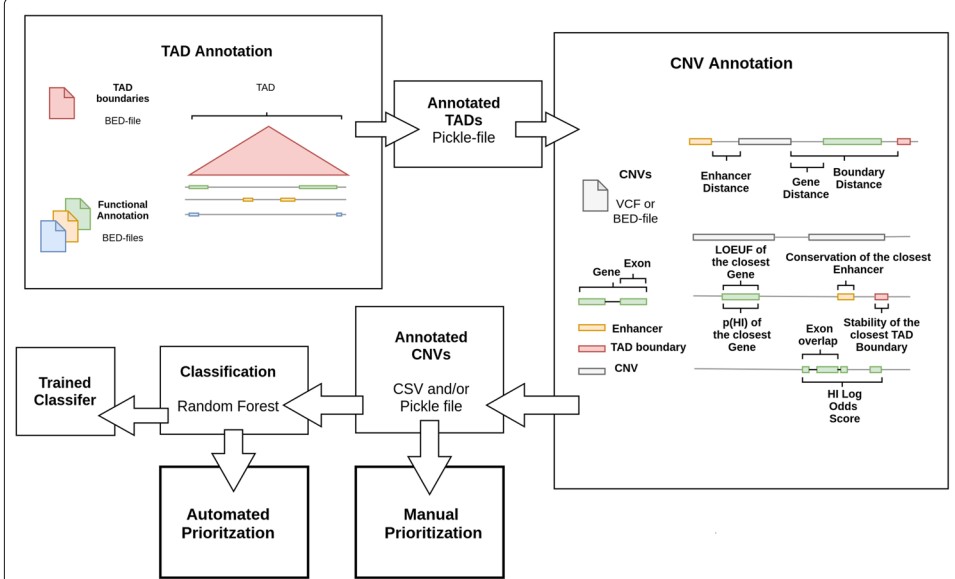

**Fig. 2** Generalised Workflow of the TADA tool. The basis for the CNV annotation are BED-files of TAD boundaries and additional sets of genomic annotations e.g. gene coordinates. In a first step, the annotation sets are sorted into the corresponding TAD environment based on genomic position. The resulting annotated TAD regions are used as a proxy of the regulatory environment during the CNV annotation ("TAD-aware annotation"). The default feature set for the CNV annotation process consists of features describing the distance to genomic elements such as genes and enhancers in the same TAD environment as well as metrics describing the functional relevance, e.g. conservation scores of affected coding or regulatory elements. Alternatively, the user can provide a set of BED-files containing the coordinates of genomic elements from which a new feature set i.e. the distance of CNVs to these annotations is generated. The user is then able to manually prioritise CNVs based on the distance features. If the default feature set is used TADA also allows for automated prioritisation using the pathogenicity score computed by our pre-trained random forest model

deletions and duplications and train separate random forest classifiers on a total of 14 functional annotation derived features (Additional file 1: Table S1). The features include: distance to the closest gene, FANTOM5 enhancer, CTCF binding site and TAD boundary in the corresponding TAD environment. Additionally, we include loss-of-function observed/expected upper bound fraction (LOEUF) [14] and Haploinsufficiency potential [4] for the closest gene as well as evolutionary conservation of the closest enhancer. Finally, we use the HI log Odds score [4] and a feature corresponding to the overlap of a CNV with potential regulatory regions based on pcHi-C [36]. Even though TADs have been shown to be broadly conserved across tissues [37], our choice of TAD boundaries identified in human embryonic tissue could still, in principle, influence the classification. To account for any cross-tissue variability, we also include a score expressing the conservation of TAD boundaries across tissues i.e. the TAD stability of the closest TAD boundary [38] as a feature. For each variant type, we split our original data into training and test set (70/30) stratified by label distribution and trained a random forest classifier. We then evaluate the performance of the parameter-tuned classifiers using three criteria 1) 5-fold cross-validation (*5-CV* set), 2) the test-set split of our original data (*Test-Split* set) and 3) on a set of pathogenic and benign *ClinVar* deletions and duplications without overlap to our training data (*ClinVar* set). This results in ROC-AUC values for the deletion model of 0.8379 (*5-CV*), 0.8059 (*Test-Split*) and 0.8865 (*ClinVar*). The ROC-AUC values for the duplication model are 0.8069 (*5-CV*), 0.7868 (*Test-Split*) and 0.8424 (*ClinVar*) (Fig. 3C). We compare our predictive performance to SVFX [22], a recently proposed alternative machine learning approach aimed predominantly at the classification of somatic CNVs. Similar to TADA, the SVFX framework allows the user to train random forest classifiers on individual variant sets to identify pathogenic potential with regards to a specific disease context. While TADA is trained on size-matched data, SVFX employs a normalisation method to account for the size bias between pathogenic and non-pathogenic variants. However, there are practical limitations to this method leading to overestimated performance metrics (see Methods for details). Thus, to allow for a sensible performance comparison, we trained a SVFX classifier on the same size-matched training data used for the TADA classifier and compared the CV performance during training as well as AUC values on the *Test-Split* and *ClinVar* variants. The SVFX 5-CV ROC-AUC values are 0.7836 and 0.7613 for deletions and duplications, respectively and the performance of SVFX on the test variants is shown in Fig. 3C. TADA outperforms SVFX for both CV ROC-AUC, *Test-Split* and *ClinVar* variants. We also compare our performance to SVScore [20], applied to the *Test-Split* and *ClinVar* variants with default parameters (see Methods for details). Briefly, SVScore calculates the mean of the ten highest CADD scores [21] in the interval affected by a CNV. This results in ROC-AUC values for the deletions of 0.6909 (*Test-Split*) and 0.8771 (*ClinVar*). For duplications, the ROC-AUC values are 0.7079 (*Test-Split*) and 0.8582 (*ClinVar*). TADA outperforms SVScore on the *Test-Split* of our original data. The difference on *ClinVar* variants is less pronounced. For *ClinVar* duplications SVScore performs marginally better than TADA. We reasoned that the increased performance of SVScore for *ClinVar* variants could be due to the different size distributions of pathogenic and non-pathogenic variants in the available training sets at the time of writing. SVScore assigns a score of 100 to all variants larger than 1 Mb, effectively labelling them as pathogenic. Hence, the method performs particularly well if most of the pathogenic variants are larger than 1 Mb and most non-pathogenic variants

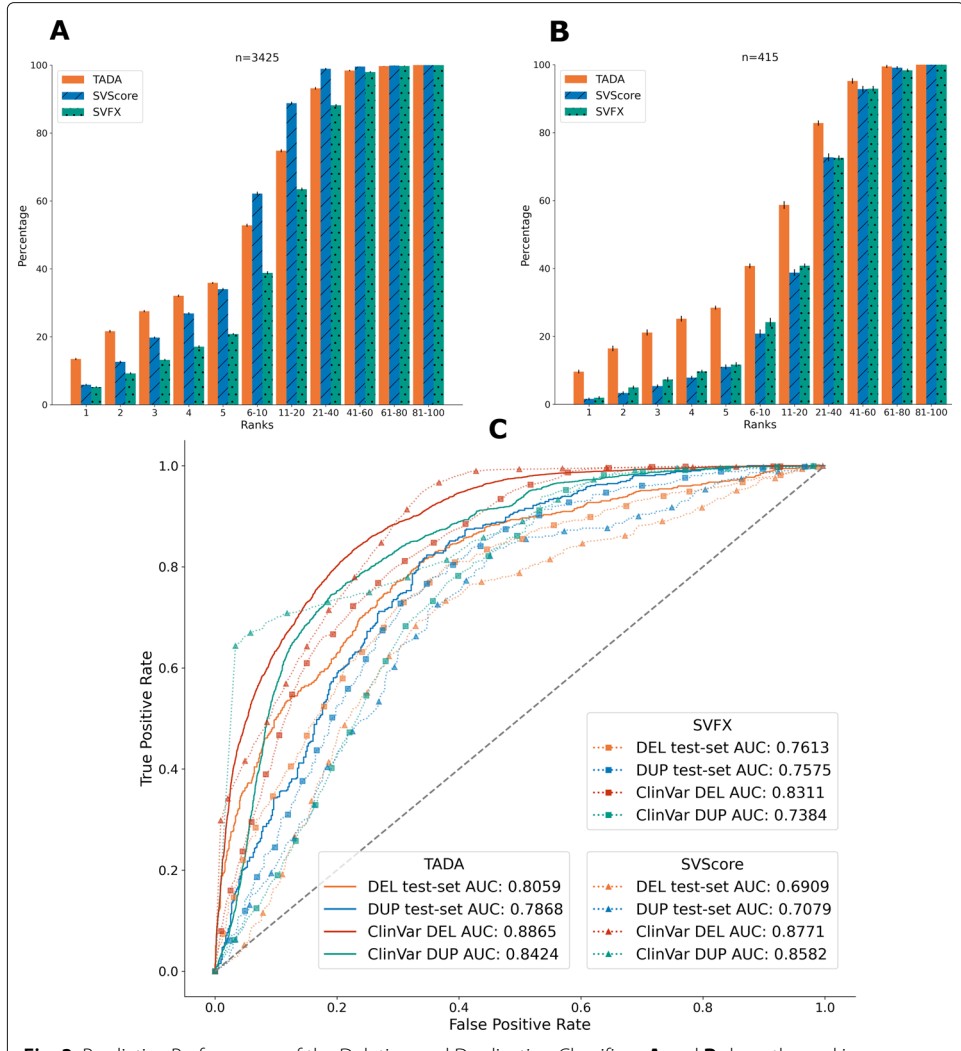

**Fig. 3** Predictive Performance of the Deletion and Duplication Classifiers. **A** and **B** show the ranking performance of TADA and SVScore for deletions and duplications, respectively. For each bin we computed the percentage of variants placed amongst the corresponding rank or ranks. Black bars indicate the standard variation based on 30 random sampling runs. **C** shows the ROC-Curves and AUC values for the deletion and duplication classifiers based on the *Test-Split* and *ClinVar* variants for both TADA and SVScore

are smaller. This likely leads to an underestimation of the pathogenic potential of smaller pathogenic variants and a likely high false positive rate for large non-pathogenic variants. To assess the effect of variant size on the classification performance, we set out to perform a second comparison between TADA, SVScore and SVFX using *ClinVar* deletions separated by size. The results (Additional file 1: Fig. S6) show superior performance of TADA across all size groups with the exception of large (> 1 mb) deletions, suggesting that TADA relies to a lesser degree than the other methods on the size difference between pathogenic and non-pathogenic variants as a discriminating feature.

Kleinert et al. recently introduced *CADD-SV* an adaptation of the original CADD method aimed at the prioritisation of SVs [39]. We show the results of a comparison between TADA and CADD-SV in Fig. S7 (Additional file 1). TADA outperforms CADD-SV with regards to the classification of both *ClinVar* and *Test-Split* CNVs. However, in

comparison to the ROC-AUC difference for *Test-Set* deletions (0.0889), the difference in performance for *Test-Split* duplications is less pronounced (0.0012).

We also compared the classification performance of TADA to a fourth alternative method, the Ensembl Variant Effect Predictor (VEP) [19]. VEP allows to annotate CNVs with a large variety of regulatory annotation and returns an *IMPACT* rating which depends on the calculated variant consequences. Since the rating is a categorisation into four groups (*HIGH, MODERATE, LOW* and *MODIFIER*) a comparison with the continuous scores of TADA and SVScore based on ROC-curves or ranking ability is not possible. We therefore computed the macro-averaged F1 scores for the test split of our original data and both the *ClinVar* variants and those smaller than 1Mb using individual thresholds for TADA, SVScore and VEP (see Methods for details). The results are shown in Table S2 (Additional file 1). TADA outperforms SVScore and VEP for both deletions and duplications of our *Test-Split* variants and *ClinVar* deletions. For the entire set of *ClinVar* duplications SVScore classifies 84% of the variants correctly which is the best macro-averaged F1 score amongst all tools. However, as shown in the last column of Table S2 and previously discussed in terms of SVScore's classification performance, the increase can be explained by the dependency on the size difference between pathogenic and non-pathogenic variants. For variants smaller than 1 Mb there is only a marginal performance difference between the tools.

Our analysis of the predictive performance across multiple test sets provides an indication of the classifier's ability to distinguish between pathogenic and non-pathogenic variants given a hard threshold on the pathogenicity probability, i.e. the probability of the variant to belong to the pathogenic class. In clinical practice the challenge is to single out a pathogenic variants from a large background of non-pathogenic variants. An ideal classifier would therefore focus on producing class probabilities, i.e. a pathogenicity score that allows to rank the true pathogenic variant as high as possible reducing the total number of variants investigated in further downstream analysis to a manageable number. As described by Benevenuta et al. [40] a potential criterion for the reliability of a classifiers class probabilities is its calibration - the fraction of true positives compared to the mean predicted value [41]. We visualise the calibration of our classifiers in Fig. S8 (Additional file 1) and further set out to evaluate the reliability of our pathogenicity score by performing a *ranking analysis*. We generate size-matched batches of 100 deletions or duplications, containing a single pathogenic and 99 benign *ClinVar* variants (Methods for details). For each batch, we computed the rank of the true pathogenic variant based on predicted pathogenicity (TADA and SVFX) as well as aggregated CADD Score (SVScore) and obtained the standard deviation of individual ranks. The ranking performance of our classifiers, SVFX and SVScore for both deletions and duplications is shown in Fig. 3A/B. Based on 3425 batches, the TADA deletion classifier outperforms SVFX and SVScore, with 35.9% correct instances in the top 5 ranks compared to 20.9% and 33.9%, respectively. However, SVScore places more correct CNVs in the ranks below the 10th rank. For duplications 28.66%, 11.7% and 11.06% of variants from 415 batches are placed amongst the first 5 ranks by TADA, SVFX and SVScore, respectively. In the duplication ranking analysis TADA outperforms SVFX and SVScore with respect to all ranks (Fig. 3B).

In analysis pipelines of sequencing results, aimed at the identification of putative pathogenic variants, it is common to discard any variants that occur frequently in a healthy population panel. A public resource may be used for this purpose, e.g. GnomAD

[14]. This process results in a set of rare variants that are more likely to be pathogenic but also more likely to be artefactual given the low observed population frequency in the reference panel. Hence, a reoccurring scenario is the prioritisation of likely pathogenic variant candidates amongst the selected rare variants. To test the performance of TADA and SVScore with regards to rare variants rather than variants classified as benign we repeated the ranking analysis comparing pathogenic *ClinVar* variants to rare ($< 0.1$ allele frequency) GnomAD variants. The results are shown in Fig. S9 and S10 (Additional file 1). In this analysis SVScore places on average 4.6% more pathogenic variants on the first 5 ranks when compared to TADA. We manually inspected individual CNVs and found that SVScore assigns low scores to all variants not directly affecting coding regions of genes while TADA also considers non-coding variants as potentially pathogenic, if the regulatory environment warrants this classification. To assess the effect of TADA's non-coding features on the ranking performance with rare variants we repeated the analysis using a modified classifier which actively penalises variants not directly affecting coding regions (Methods for details). The resulting ranking performance is shown in Fig. S11 and S12 (Additional file 1). The performance of TADA improves compared to the initial ranking analysis of rare variants. However, we reasoned that a model solely focusing on coding regions would not reflect currently emerging and future CNV effects well as multiple instances of non-coding pathogenic variation are being identified [5, 42]. The presented and published model therefore also includes features accounting for the CNV's non-coding regulatory environment.

We further tested the performance of our pre-trained duplication model on variants of two developmental disease (DD) patients reported in a previous study focused on the benefit of Hi-C for SV identification [43]. In this analysis, we applied our model trained on DECIPHER data on duplications identified in the individual DD2, including the coordinates of the disease-causing duplication of DD1 (see Methods for details). The computed pathogenicity scores of the 255 duplications including the pathogenic variants of DD1 and DD2 are shown in Fig. S13 (Additional file 1). Both disease-causing duplications were assigned pathogenicity scores higher than the 90th percentile (0.4336). The disease-causing DD2 duplication was placed on rank 2 amongst the 255 variants with a pathogenicity score of 0.7986. The disease-causing DD1 duplication was ranked notably lower with a score of 0.5865. To investigate the regulatory environment at the duplication loci as represented in our feature set we visualised the corresponding loci using IGV [44] as shown in Fig. S14 (Additional file 1). Both variants overlap with TAD boundaries, several gene as well as enhancers annotations. The DD2 duplication directly compromises the SOX9 locus—a highly haploinsufficient DDG2P gene. As presented in the following paragraph and shown in Fig. S15 (Additional file 1), TADA's classification process for duplications is highly influenced by the HI-Score of the closest gene. Thus, The overlap with the SOX9 likely drives the high pathogenicity score predicted for the DD2 duplication. The DD1 duplication—as suggested by Melo et al.—effects the expression of KCNJ2 through the formation of a new chromatin domain (neo-TAD) that includes copies of KCNJ2 and KCNJ16 as well as SOX9 enhancers likely causing the patients phenotype. While TADA does recognise the close distance of this duplication to developmentally significant genes defined by the HI score feature and its overlap with TAD boundaries as well as enhancers at this locus, it likely does not recognise the complex pathogenic effect through the neo-TAD formation resulting in a decreased pathogenicity score.

Our classifier performs well on developmentally associated pathogenic variants and shows a high test-set and ranking performance for the *ClinVar* database, indicating that the model can be applied to a disease and sample context unrelated to the training set. However, both DECIPHER and *ClinVar* include primarily coding sequence affecting variants due to the limitations of the underlying experimental procedures. Only since the inclusion of WGS—especially long-read WGS—in clinical practice, the focus on non-coding disease-causing CNVs has increased leading to more comprehensive catalogues of annotated pathogenic variants. Still, the current publicly available call sets include very few non-coding sequence affecting disease-causing variants which is likely to be reflected in our trained classifier. We therefore decided to determine the most relevant features and identify potential biases in our trained model. To account for any biases introduced by correlated features, we computed the partial correlation between our features (Additional file 1: Fig. S16), and computed the mean loss in accuracy for partial correlation clusters after permutation of highly correlated feature clusters [45] (Methods for details). The mean loss and standard deviation across 30 computations with different random seeds is shown in Fig. 4 and Fig. S15 (Additional file 1) for deletions and duplications, respectively. As expected, the results indicate that both the deletion and the duplication model rely mainly on coding rather than non-coding functional annotation. The most relevant features for the trained classifiers are the predicted haploinsufficiency of the closest gene and the HI Log Odds score, followed by the distance to the closest DDG2P

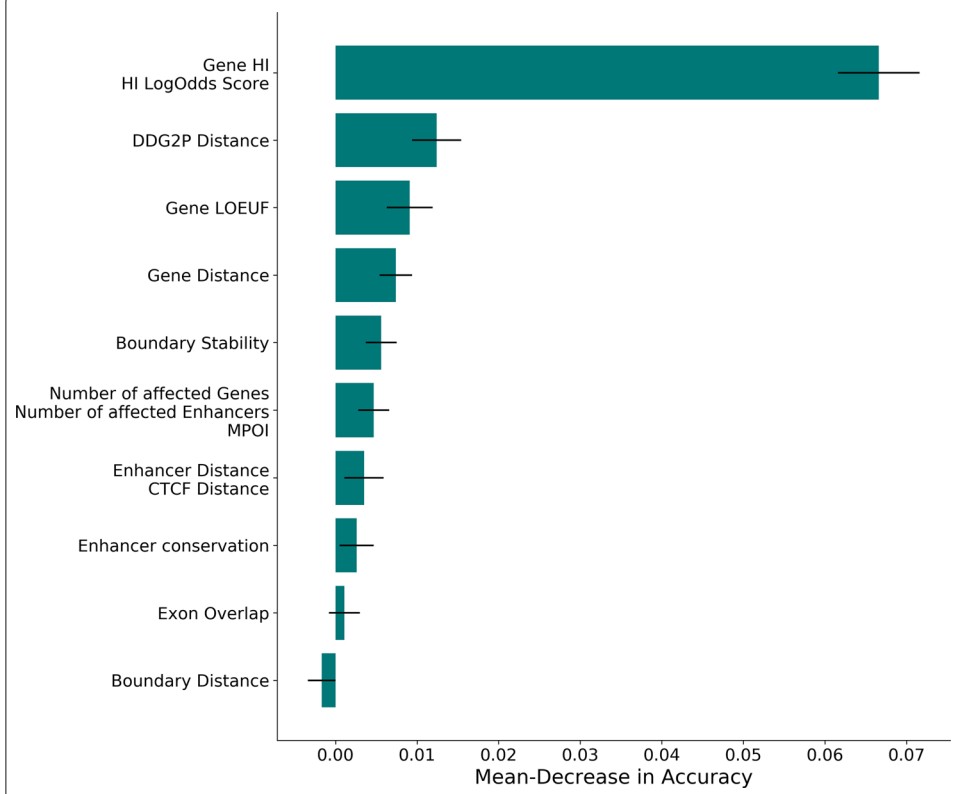

**Fig. 4** Feature Importance of the Deletion Model. The figure shows the mean loss in accuracy after permutation of highly correlated feature clusters (see Methods for a detailed description of individual features). The standard deviation based on 30 sampling runs with variable random seed is indicated by black lines

gene. Regulatory annotations are of lower importance for classification. This is in agreement with our enrichment results, where we could not observe enrichment of pathogenic DECIPHER variants in FANTOM5 enhancer annotations, and our previous comparison of ranking ability for rare variants between TADA and SVScore. However, we expect non-coding annotation to gain importance as potential pathogenic non-coding training variants increase.

## Discussion

The prioritisation and identification of disease-causing genomic variants is an active field of genomic research. SNPs and InDels and their relation to human disease have been the focus of intense study [1, 2]. Although early evidence suggests SVs are a major contributor to genome-wide variation, comparatively little is known about their disease impact at an individual and population level. This can be attributed to technical limitations, namely the accurate identification of SVs and precise SV breakpoint calling. It follows that few studies have, so far, focused on the prioritisation and ranking of large sets of SV calls to highlight smaller and more manageable subsets of SVs with higher predicted disease potential.

Experimental as well as algorithmic advancements in the field of SV detection are leading to an increase in publicly available catalogues of SVs [14, 15]. These catalogues are focused mainly on common, i.e. likely non-pathogenic variants and reveal patterns of depletion in coding and regulatory functional annotation. Due to the lack of publicly available data, an analysis on a comparatively comprehensive catalogue of pathogenic SVs is missing.

In this work, we used the DECIPHER database of pathogenic CNVs to analyse the potential of functional annotation and CNV overlap as a distinguishing feature between pathogenic and non-pathogenic CNVs. We conducted an extensive series of enrichment tests, identifying contrasting patterns of enrichment between pathogenic and non-pathogenic CNVs for multiple sets of genomic annotation. We observed significant depletion of non-pathogenic variants in coding and regulatory regions, positively correlated with the intolerance to LoF-mutations and predicted haploinsufficiency of coding regions as well as the primary sequence conservation of regulatory regions. In our analyses we observed a contrasting pattern of enrichment for pathogenic variants in coding regions and CTCF binding sites, providing additional evidence for the potential of functional annotation as a feature to identify pathogenic variants. In further enrichment analyses we included ChromHMM annotations, SDs and HARs. We were not able to observe a contrasting pattern of enrichment and depletion in this group of annotations, suggesting that ChromHMM annotations, SDs and HARs do not represent genomic regions with differential selective pressure towards perturbating variation. We also investigated the overlap of pathogenic and non-pathogenic CNVs with TAD-centric features, i.e. TADs stratified by regulatory importance. We observed a significant depletion of non-pathogenic variants in TADs containing regulatory and coding annotation. In contrast, pathogenic variants are significantly enriched in TADs of regulatory importance. This could suggest that the selection towards variation affecting functional annotation is likely to extend to entire regulatory domains. However, to confirm this observation further analysis needs to be performed with a more extensive catalogue of TAD boundaries across tissues as well as variants, less liable to the investigator bias towards the coding sequence effect of CNVs.

The enrichment analyses also revealed significant depletion of pathogenic variants in TAD boundaries and highly conserved enhancers, indicating that regulatory functional annotation is less important than coding sequence-centric annotation in aiding the discrimination of pathogenic versus non-pathogenic variants based on the data sets used in these analyses. This is perhaps unexpected, given there is increasing evidence on the role of variants impacting non-coding regulatory elements on rare disease [5, 33]. Accounting for the focus of DECIPHER on the coding rather than non-coding effect of CNVs, we reasoned that the significant depletion of pathogenic variants in non-coding functional associated annotation is a consequence of investigator bias in our set of annotated pathogenic variants. We anticipate that, with larger less biassed pathogenic SV repositories becoming available, observed genome-wide SV impact on regulation will yield a stronger signal. Current prioritisation methods should therefore provide the flexibility to account for both coding and non-coding effects.

For this purpose we developed TADA, a method to prioritise pathogenic CNVs based on their overlap with functional annotation. The tool provides the option to manually *prioritise* variants, i.e. it returns annotated CNVs as a list that can be sorted by each of the annotations. It also allows for machine learning-based prioritisation using a random-forest model trained on a functional annotation-based feature set. The default feature set that we provide includes coding and non-coding associated features, motivated by the results of our enrichment analysis. Alternatively, the user can provide a set of genomic annotation associated with a disease or sample context to generate a custom feature set. Even though, based on the enrichment analysis, non-coding features such as TAD boundaries and enhancer conservation do not assist in the differentiation of pathogenic from non-pathogenic variants, we decided to include them in the default feature set. We argue that during manual annotation an prioritisation based on the default feature set the non-coding can prove to be beneficial in identifying pathogenic variants. We trained random forest classifiers for deletions and duplications and found they performed well on the *Test-Split* and *ClinVar* variants, outperforming SVFX and CADD-SV with regards to all test variants and SVScore as well as VEP on all test data with the exception of *ClinVar* duplications. In a second analysis using *ClinVar* deletions grouped by size, TADA shows superior performance for smaller to medium sized variants, indicating the robustness of our model towards the size difference between pathogenic and non-pathogenic variants. To simulate the application of our models in a clinical setting, we computed the precision of the pathogenicity score over batches of benign variants combined with a single pathogenic variant. Both the deletion and duplication model performed well, placing more than 30% of the pathogenic variants amongst the first rank based on the pathogenicity score, again outperforming SVFX and SVScore. In an analysis focused on two DD-patients, TADA assigned high pathogenicity scores to both disease-causing variants, ranking the pathogenic duplications for the DD2 individual 2nd amongst 255 variants. The above indicates that the pathogenicity score is a close approximation of true pathogenic effect in our test set—especially given the superior ranking ability of our models. As expected, the analyses of the classifier's feature importance revealed dependency on coding regions rather than regulatory regions, mirroring the results of our enrichment analyses. We therefore recommend the application of the automated prioritisation using the pre-trained random forest model with focus on the coding rather than non-coding effect of CNVs. Since TADA is trained on the DECIPHER variants, which were identified

as the cause of developmentally associated disease phenotypes, we cannot guarantee that the pre-trained model is able to accurately classify variants in a different disease context. Hence, we provide the possibility to manually prioritise variants based on a user-defined or on the default feature set, which also includes features accounting for the non-coding effect of CNVs.

## Conclusions

Thanks to recent experimental and algorithmic approaches SVs can be reliably identified and their role in clinical diagnostics is beginning to be established. This raises the need for methods to assist in the identification of pathogenic SVs. Still, the proportion of balanced SVs in publicly available databases of pathogenic variation is comparatively low, limiting any supervised machine learning approach focusing on the prioritisation of pathogenic variants, including TADA. to unbalanced SVs. Nevertheless, our results show that the automated prioritisation of pathogenic CNVs based on functional annotation is a promising approach. The TADA framework outperforms several comparable methods with respect to overall predictive performance and ranking ability. With the likely increasing number of more comprehensive available variant catalogues, we aim to improve the predictive power of our classifier and adapt our approach to other classes of disease-relevant genomic structural variation.

## Methods

### Variant sets

The number of deletions and duplications in the individual data sets pre- and post-filtering as well the merged common call set are shown in Fig. S17 (Additional file 1).

We obtained 31,615 CNVs from DECIPHER [23]. We filtered the CNVs according to their pathogenicity and size, and chose to retain variants categorised as *pathogenic*, *likely pathogenic* or *unknown* with a size larger than 50 bp (20,984 remaining). Since DECIPHER serves as database to analyse candidate i.e. potential disease-causing variants, we reasoned that a large proportion of variants with *unknown* effect are still likely pathogenic. We noticed that multiple DECIPHER calls were overlapping and possibly representing the same variant, we selected the smallest variant for each pair/cluster of overlapping variants based on a 90% reciprocal overlap (18,792 remaining). Finally, we only kept variants located on autosomes (18,786 remaining).

The common i.e. *non-pathogenic* variant set is a compendium built from four different data sources: a publication by the Eichler group featuring SVs identified via deep PacBio sequencing of 15 individuals [15] (97,585 variants), a collection of 14,891 individuals published by the GnomAD consortium [14] (445,858 variants), a set of CNVs called from the UK Biobank data set [25] (275,180 variants) and CNVs obtained from the Database of Genomic Variants (DGV) [24] (114,555 variants). Variants in the set published by Audano et al. [15] were mapped to GRCh37 using the *LiftOver* tool [46]. Three thousand seven hundred forty-seven of 97,585 variants were lost during this process. All other SVs were already mapped to GRCh37. In order to match the pathogenic variants we only kept non-pathogenic CNVs located on autosomes larger than 50*bp*. We also discarded rare and potentially deleterious variants by applying individual filters to each of the data sources: we filtered for *Shared* or *Major* deletions published by Audano et al. for i.e. variants present in all or ≥ 50% of the samples (5404 remaining), GnomAD SV variants

with an allele frequency (AF) > 0.1 (5037 deletions and 1494 duplications remaining), UK Biobank deletions supported by 3 or more samples (33,373 remaining) and DGV variants reported in more than one publication (27,235 deletions and 6138 duplications remaining). To account for overlapping variants between different sources of non-pathogenic variants, we clustered variants with a reciprocal overlap greater or equal to 90% . For each pair/cluster of overlapping variants we selected a single variant based on their origin, with the following prioritisation order: Audano et al. 2019, GnomAD-SV, DGV and UK Biobank. We reasoned that variant calls reported based on sequencing—especially long-read-sequencing—provide more precisely resolved breakpoints than variants reported based on array-CGH or SNP-arrays. This resulted in a total number of 53,003 deletions and 7606 duplications.

While the pathogenic variants were called using array-CGH, the GnomAD and Audano variants are based on WGS and long-read-sequencing, respectively. The difference in experimental methods is reflected in the size distribution of pathogenic and non-pathogenic variants (Additional file 1: Fig. S18). To account for the size bias across variants sets, we binned the non-pathogenic variants by size using an empirical cumulative distribution function (ECDF) with bin size 60. We then sampled for each bin the same number of pathogenic variants. For bins with a higher number of non-pathogenic than pathogenic variants, we sampled the same number of non-pathogenic variants as pathogenic variants without replacement. The final set of deletions used for training and testing consisted of 6130 CNVs, 3065 pathogenic and 3065 non-pathogenic matched by size. The final set of duplications included 3410 CNVs i.e. 1705 pathogenic and 1705 non-pathogenic size-matched variants, also split 70/30 into training and test sets. The proportion of non-pathogenic variants by data source changed during the size-matching, due to the lack of short pathogenic variants. Figure S19 (Additional file 1) shows the proportion of variants by pathogenicity and data source before and after the size matching. Figure S20 (Additional file 1) shows the distribution of non-pathogenic CNVs by data source across the genome.

We obtained *ClinVar* CNVs from (https://www.ncbi.nlm.nih.gov/clinvar/) on October 24, 2019 restricting the search by *Type of variation* to *copy number gain*, *copy number loss*, *deletions* and *duplications*. First, we stratified the variants by type of variant i.e. deletions and duplications (73,533 deletions; 47,022 duplications). Then, we separated both deletions and duplications into pathogenic (*Pathogenic* and *Likely pathogenic* annotation) (11,816 deletions; 3880 duplications) and benign (*Benign* and *Likely benign* annotation) (13,381 deletions; 11,609 duplications) and only kept variants located on autosomes. Finally, we discarded any duplicated variants (as described above for the DECIPHER dataset) and those overlapping with our training data (90% reciprocal overlap) in each set of pathogenic/non-pathogenic deletions and duplications (17,553 deletions; 10,062 duplications). We conducted an additional analysis to investigate the effect of the reciprocal overlap threshold on our classification performance. The results are shown in Fig. S21 (Additional file 1). With the exception of a slight increase in classification performance from 10 to 20% the AUC-ROC values remain similar across the range of reciprocal overlap thresholds.

For the analysis of the classifiers' ranking performance, we used the above described filtered *ClinVar* variants. To analyse the ranking performance for rare rather than benign *ClinVar* variants we used GnomAD CNVs with an AF <= 0.1. We filtered the rare CNVs

for duplicated variants, discarding any rare CNVs overlapping with our training data (90% reciprocal overlap). This resulted in a variant set comprised of 46,880 duplications and 167,817 deletions.

### Annotations

We obtained hg19 TAD boundaries from human embryonic cells [6] as described by MCArthur et al. 2020 [38] annotated with *boundary stability*. For both the enrichment testing and the training of the pathogenicity classifier we used FANTOM5 enhancer annotations [47] without any tissue specifications. We annotated the enhancers using aggregated 100-way PhastCons [48] i.e. the mean of base-wise conservation scores over the enhancer interval. The *conserved* and *highly conserved* enhancers in the enrichment analysis correspond to FANTOM5 enhancers with aggregated PhastCons scores over the 75% and 90% percentiles given the background distribution over all enhancer annotations. We obtained gene annotations from the GnomAD consortium and exon annotations for the computation of the exon overlap feature from GENCODE (comprehensive gene annotation v.32) [49]. We stratified the gene annotations by predicted Haploinsufficiency ($p(HI)$) [4] and intolerance to loss-of-function mutations i.e. LOEUF [50]. For the enrichment analysis we defined genes with $p(HI) > 0.9$ and $p(HI) < 0.1$ as *HI Genes* and *HS Genes*, repsecitvly. Genes with LOEUF< 0.1 and LOEUF> 0.9 as *ploF Tolerant Genes* and *ploF Intolerant Genes*. We obtained hg19 CTCF binding site annotations from ENCODE i.e. irreproducible discovery rate (IDR) optimal ChIP-seq peaks (ENCODE Accession Number: ENCSR000EFI). The computation of overlap with potential regulatory regions identified in pcHi-C data is based on data from [36]. We selected the *P-O-interactions* ($-log10$(p-value) $\geq$ 3) for each gene contained in our set of annotated genes and computed the overlap of a CNVs with each interacting fragment i.e. 1 if the CNV overlaps with a fragment, 0 otherwise. Finally, we divided the sum over all interacting fragment for each gene by the genes LOEUF value. We obtained hg19 telomeric regions from the UCSC genome browser [46] and extended them by 5 mbp to match the annotations described by Audano et al. 2019 [15].

### TADA workflow

During the TADA's prioritisation process we first perform a *TAD-aware Annotation* by computing the overlap of TADs with regulatory elements, namely genes, enhancers and CTCF sites, and collect the overlapping annotation for each TAD region. This preprocessing step provides two advantages in the subsequent CNV anntotation. First, it allows for a faster annotation of CNV regions, since all features only have to be computed with regards to the regulatory elements in the same TAD environment of a CNV e.g. not all genes have to be considered during overlap computation. Secondly, features extending the genomic sequence directly affecting the CNV such as the gene associated with overlapping interacting fragments (derived from pcHi-C) are restricted by the TAD environment and, therefore, potentially reflect the true regulatory impact of a CNV better. We also allow for a user-defined sets of genomic elements during the Annotation process. In this case the user needs to provide coordinates in BED-files of genomic elements which are then again sorted into their corresponding TAD environment. The TAD environment can also be provided by the user. During the CNV annotation the distance of CNVs to their respective closest element of each set of genomic elements is used as a

feature. The user is then allowed to either train a classifier on two datasets using their own feature set or manually prioritise variants based on the distance features. Alternatively, we provide the default feature set as described previously for manual and automated prioritisation.

### Enrichment testing

We performed the enrichment tests using the *gat-run* protocol of the *Genome Association Tester* (GAT) [29]. GAT is a bootstrap-based approach used to test the association between sets of genomic intervals. The *gat-run.py* protocol merges the segments of interest i.e. the CNVs in a preprocessing step, which resulted at first in a coverage (track density) above 90% for each variant set. To avoid false positive enrichment due to size bias, we therefore used the size-matched CNVs sets and reduced the track density for pathogenic and non-pathogenic variants to 30.1% and 22.7%, respectively.

CNVs are non-randomly distributed across the genome. Regions with an increased amount of paralogous repeats such as segmental duplications are prone to non-allelic homologous recombination, a recombination process that can lead to the formation of deletions or duplications [51]. To account for the non-random distribution of CNVs, we used GC-content families [26] to split the genome into *isochores*. For each isochore we performed a separate enrichment test using the *isochores* function of GAT. We used *gat-run.py* with the following settings: *–nbuckets 10,000 –bucket-size=960 –num-samples=10,000 –counter=segment-overlap*. To compare the FCs between pathogenic and non-pathogenic variants we used *gat-compare.py*.

### Classification and Preprocessing

We annotated the CNVs with a total of 14 features. In preparation to the classification we split the data into training and test set (70/30 split). We then imputed missing data in both sets with the mean of the corresponding feature in the training set. To account for the differences in data ranges of raw feature values and decrease the convergence time during training, we scaled all features to a range between 0 and 1. Similar to the imputation process, we fit the scaler on the training data and applied it to the test data. Finally, we trained a Random Forest on the imputed and scaled training set. We then evaluated the performance based on stratified 5-fold cross-validation, on the separate test set and on *ClinVar* variants using AUC. We also computed the individual performance of our model pathogenic and non-pathogenic *Test-Split* variants using the F1-score. This resulted in values of 0.73 (non-pathogenic) and 0.75 (pathogenic) for our deletion model and 0.71 (non-pathogenic) and 0.75 for the duplication model.

As a comparison to our predictive performance we trained an individual model on our training data using the SVFX framework as instructed on https://github.com/gersteinlab/SVFX. The SVFX authors suggest to use a normalisation preprocessing step to avoid potential size bias between pathogenic and non-pathogenic variants. However, a closer inspection of the underlying code revealed a z-score normalisation applied to the entire data set before CV i.e. splitting the data into training and test set which directly influences the CV measured performance during training. Since our training set has been previously size-matched and this normalisation step outside of the CV-loop would cause information leakage, we used SVFX without prior z-score normalisation of features. We then applied the trained model to predict the pathogenicity of the individual test-sets i.e. the

test set split and *ClinVar* variants. We also scored the test variants using SVScore (v. 0.6) with default parameters i.e. *top10weighted* operation and then normalised the scores for each set of variants between 0 and 1 to allow comparison to the pathogenicity scores computed by TADA. To compare our method to VEP we used a conda environment (https://anaconda.org/bioconda/ensembl-vep) v.96.0 and applied the *vep* function with the default GRCh37 annotations (Ensembl database version 102) and $-max\_sv\_size$ set to $10^9$ on the seperate test set and *ClinVar* variants. We labelled variants with IMPACT rating *HIGH* or *MODERATE* as pathogenic and computed the macro-averaged F1 score for each variant set using the *f1_score* function of *sklearn*. Similarly, we computed F1 scores for TADA and SVScore across all variants sets. For TADA we classified all variants with a pathogenicity score higher than 0.5 as pathogenic and for SVScore we used the 90th percentile of each CNV set to distinguish between pathogenic and non-pathogenic variants as this is the threshold with the highest recorded performance in the authors analysis [20]. We installed CADD-SV (v.1.0) locally as described at https://github.com/kircherlab/CADD-SV. Since CADD-SV requires SVs to be mapped to the GRCh38 reference, we used the *LiftOver* tool to retrieve GRCh38 coordinates for all variants used in the performance comparison. In the *LiftOver* process 80 *Test-Split* deletions, 62 *Test-Split* duplications, 255 *ClinVar* deletions and 404 *ClinVar* duplications were lost. As described in the CADD-SV manuscript we used the maximum of *span* and *flank* raw scores as an indicator of CNV pathogenicity and additionally employed a min-max-normalisation for each variant set to allow for the performance comparison with TADA.

### Ranking performance

In order to test the ranking performance of our trained model i.e. its ability to differentiate the true pathogenic variant from a set non-pathogenic variants we used the above described benign and pathogenic *ClinVar* variants. We predicted the pathogenicity score using our pre-trained random forest classifier and the SVFX model as well as aggregated CADD scores computed by SVScore. We then binned the rare variants by size using an ECDF with 60 bins. For each pathogenic variant we selected the closest bin of non-pathogenic variant by size discarding any variants larger than the largest non-pathogenic variants. If the corresponding bin contained 99 or more variants we randomly sampled 99 non-pathogenic variants. For the *ClinVar* variants, this resulted in a total of 3425 and 415 batches of *variants* for deletions and duplications, respectively. We then sorted the non-pathogenic variants and the single pathogenic variant by predicted pathogenicity and used the index in the sorted list as predicted rank. In order to generate standard deviations for the variant placement we repeated the sampling of non-pathogenic variants over 30 (*numpy.random.choice*) random seeds. We repeated the sorting process with the normalised scores computed by SVScore and the pathogenicity score produced by the SVFX model and again used the index of the pathogenic variant in the sorted list as rank.

In order to asses the ranking performance of TADA and SVScore for rare variants we repeated the above process with rare variants obtained from Collins et al. 2019 [14] using the initial model and modified TADA classifier where the HI, HI log odds score and LOEUF were all set to fixed values, namely 0, − 10 and 2 if a CNV does not directly affect a coding region, actively penalising non-coding CNVs. The fixed values were picked according to the maxima/minima that correspond to the least regulatory importance of each feature. For example, genes annotated with higher LOUEF scores are less

likely affected by loss-of-function mutations with 2 being the highest possible value. The number of batches used in this analysis, each including a single pathogenic and 99 rare variants, were 4308 and 586 for deletions and duplications, respectively.

For our performance analysis on DD-patients, we obtained the set of the DD2 duplications upon request from the authors of the original study [43]. The coordinates of disease-causing duplications were presented in the supplement of their publication. We filtered the set of 1260 ERDS calls for duplications and included the disease-causing DD1 variant, resulting in a set of 304 duplications of which 255 overlapped with our annotated set of TADs, allowing them to be prioritised by our duplication model trained on DECI-PHER data. To produce the screenshot visualising the regulatory environment we used IGV v11.0.10 [44].

### Feature importance

We employed hierarchical clustering using the *scipy* python package to generate clusters of highly correlated features based on the training set of annotated size-matched pathogenic and non-pathogenic CNVs. For each cluster with a maximal distance of one we permutated the correlated feature columns in our training data and computed the predicted accuracy using the pre-trained random forest model. We then reported the difference between the accuracy based on the original and permutated data set. Both accuracies are based on the out-of-bag samples of the random forest model. Using the *numpy.random.choice* function we generated 30 random seeds between 0 and 100 and repeated the permutation process for each random seed. We then computed the mean and the standard deviation for the distribution of accuracy differences of each cluster.

### Supplementary Information

---

**Additional file 1:** Figures S1-S21.
Tables S1-3.
**Additional file 2:** Review history.

---

#### Acknowledgements
We thank Verena Heinrich, Robert Schöpflin, Uirá Souto Melo, David Heller and Emel Comak for useful discussions. We also thank James Priest and Matthew Aguirre for providing us with access to the CNVs calls from the UK Biobank. This study makes use of data generated by the DECIPHER Consortium. A full list of centres who contributed to the generation of the data is available from http://decipher.sanger.ac.uk and via email from decipher@sanger.ac.uk. Funding for the DECIPHER project was provided by the Welcome Trust.

#### Peer review information

#### Review history
The review history is available as Additional file 2.

#### Authors' contributions
JH planned and performed all bioinformatic analyses and implemented the TADA software. GG devised the study design and supervised the work. JH and GG wrote the manuscript. MV and SM supervised the work and contributed to the preparation of the final manuscript. The authors read and approved the final manuscript.

#### Funding

#### Availability of data and materials
The TADA source code and documentation is available at https://github.com/jakob-he/TADA/ [52]. We provide a table containing download links and obtained dates for both the variant and annotation sets in the supplement (Additional

file 1: Table S3). The datasets generated and/or analysed during the current study with the exception of the DECIPHER variants are also available at https://github.com/jakob-he/TADA/manuscript. The version of the source code used in the manuscript is deposited on Zenodo [53].

## Declarations

### Ethics approval and consent to participate
Not applicable.

### Consent for publication
Not applicable.

### Competing interests
The authors declare that they have no competing interests.

### Author details
[1]Max Planck Institute for Molecular Genetics, Ihnestraße 63, 14195 Berlin, Germany. [2]Charité Universitätsmedizin Berlin, Charitéplatz 1, 10117 Berlin, Germany.

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

## 