## [**Additional file 2** Review history. · Genome Biology]

Review History

First round of review

Reviewer 1

Were you able to assess all statistics in the manuscript, including the appropriateness of statistical tests used? No.

Were you able to directly test the methods? No.

Comments to author:

The manuscript submitted by Hertzberg and colleagues presents a new method (TADA) for predicting the pathogenicity of Copy Number Variants. The paper address one important problem in the field. I believe that the manuscript could be considered for publication after addressing the comments reported below.

Major Revisions

1) The description of the composition of the datasets use to test TADA should be improved. The authors should include a table summarizing the number of variants in each dataset and their compositions in terms of classes.

To make the work reproducible the authors should provide as supplementary materia a file including the all variants included in each dataset with their annotation.

2) In the datasets used for trining and testing you also considered the "likely pathogenic" variants. In general this annotation is obtained using predictive methods therefore such mutations could be potentially easier to predict. Could you please report the performance of the TADA excluding such mutations from the testing set? Is there any differences in the performance?

3) In the discussion, it is mentioned that the prediction of the Haploinsufficiency is the most discriminating feature. Can you please report the performance of the method using different combinations of the 14 features to show what is gain in terms of AUC when adding less discriminating features?

4) As you have pointed out the length of the CNV is a critical feature for the predictions. What is the performance of the method if you split the training and testing sets in short and long CNVs?

5) A possible solution to mitigate the bias due to the annotation features is to perform a chromosome-based cross-validation procedure. If you keep all the variants of a chromosome in the same subset do you observe any differences in the performance?

Minor revisions

1) Many numbers in the text are reported without any digits before the decimals. I suggest to

include 0 when needed.

2) In the Annotations section at page 20 you mentioned the selected P-O-interactions were selected assuming a p-value=3.

I believe this is a typo?

3) Please specify better what does it mean aggregated PhastCons and PhyloP scores? Do you use a combination of them? What are the number of species included in the alignment used for the calculation of such scores?

Reviewer 2

Were you able to assess all statistics in the manuscript, including the appropriateness of statistical tests used? No.

Were you able to directly test the methods? No.

Comments to author:

Hertzberg et al., proposed a new method named TADA for prioritizing putative pathogenic CNVs. In principle, the authors established a random forest model using multiple genomic features including TAD boundaries, FANTOM5 enhancer annotation, genomic conservation, CTCF binding, and haploinsufficiency and intolerance of loss-of-function mutations for genes, etc. To train and test the model, they collected both non-pathogenic and pathogenic CNVs (including deletion) from DECHIPHER, ClinVar CNVs, and GnomAD. The model proposed several important features in classification, mostly associated with the pathogenic score of associated genes. However, such a result is expected due to the biased data collection between coding and non-coding associated pathogenic variants, thus it is hard to conclude that the identified features provide additional insight to annotate pathogenic CNVs. Further, the authors claimed that TADA is accurate than other existing methods and emphasized the utility of the new tool in predicting the disease impact of large genomic alterations, but the currently presented results may not be sufficient to evaluate the validity of TADA.

Major comments

- It is difficult to figure out the uniqueness of TADA compared to other benchmarked algorithms. For example, the authors demonstrated that SVFX is a classifier relying on somatic variants as a proxy for pathogenicity, but the SVFX model also generated a random-forest classifier based on data-driven approaches with various genomic features similar to TADA. Despite the slight difference in feature sets, it seems SVFX and TADA use essentially the same framework. To deliver the difference more clearly, the authors should evaluate the performance for the same data set used in SVFX and also provide specific examples and statical analysis to demonstrate how each method offers different pathogenic annotations.

- CNVs are very common in the genome, even for healthy individuals. What happens if the authors provide only non-pathogenic germline CNVs as an input? Does TADA recognize all these germline variants as non-pathogenic?

- As pointed by the authors, the method is highly biased to coding sequence associated variants due to the biased data collection. However, the mechanism of action of coding-sequence and noncoding-sequence associated variants should be different. Thus, the authors should test how the performance differs when using only coding sequence associated variants and non-coding sequence associated variants.

- To make the prediction result more interpretable, I'd like to suggest include several example genome browsers snapshots and demonstrate how the variants are recognized as pathogenic and what is the expected mechanism.

- The choice of feature data set could be important in classification performance. For example, TAD annotation is entirely arbitrary. Depending on the hard cutoff or parameters used, the TAD annotation could also identify sub-TAD structures. As exemplified in TAD annotation several genomic features are highly variable depending on the user-defined parameters, the authors should show how such variations affect the robustness of TADA.

- Page 13 line 391 'significant depletion of pathogenic variants in TAD boundaries~': such interpretation should be biased by the over-collected coding-sequence-associated pathogenic variants.

- As shown in Figure 4, all essential features are associated with pathogenic impact on involved genes. This result might imply that the essential part of TADA classification is known pathogenicity of the corresponding gene disruption rather than other genomic features. Suppose these genomic features such as TAD boundary, CTCF distance, enhancer distance are not critical in TADA. In that case, it is unclear the advantage of TADA in classifying pathogenic variants rather than simple classification based on the pathogenic effect of affected genes.

Reviewer 3

Were you able to assess all statistics in the manuscript, including the appropriateness of statistical tests used? Yes.

Were you able to directly test the methods? No.

Comments to author:

Hertzberg et al. present the new tool "TADA" for the manual or machine learning-assisted prioritization of unbalanced SVs (CNVs; deletions and duplications) in terms of their pathogenic potential. The authors perform an extensive enrichment analysis to identify genomic annotation categories (genes, enhancers etc., and restricted by the local regulatory environment defined by TAD boundaries) that are associated with CNV pathogenicity. The identified annotations are then used to inform feature design for the machine learning part of TADA. The classification models are trained and evaluated on different sets of curated CNVs and seem to deliver a more stable and often better performance than competing approaches. The authors acknowledge that there are several factors that likely impact the general applicability of TADA (e.g., detection and annotation biases for the curated CNVs) and that are hard to quantify. Overall, TADA is a thoroughly designed and tested tool that tackles an important problem. Since a main output of the (automated) TADA analysis is a ranked list of potentially pathogenic CNVs, its output needs further evaluation by a domain expert.

Below is a list of more specific comments:

Comments:

2/41: I am unclear about your criticism of the SVFX tool. Is that speculation, or do you substantiate that claim somewhere?

3/56 "biological relevance": your use of the expression "biological relevance" seems a bit overly assertive in the given context. The enrichment tests suggest an association, but the actual

relevance can commonly only be assessed in the wet lab.

4/85: any specific reason why you did not use the large set of SVs published a couple of months ago in Science ("Haplotype-resolved diverse human genomes and integrated analysis of structural variation") when compiling your data set?

4/106: please make explicit what quantity the q-value is designating (re: multiple testing/FDR).

5/135: please be more verbose on why you think the observed depletion at TAD boundaries is "interesting".

6/140ff: I have trouble understanding why you have that break before adding HAR, ChromHMM and SD annotations to your enrichment analysis? None of those categories seem far-fetched when looking for potential associations to CNVs. Also, it is hard to imagine that any set of current genomic annotations would verifiably represent the full (!) spectrum of regulatory activity. Maybe you make a distinction here that I have missed?

7/192ff: indeed; can you summarize what potential (annotation) biases could be in your set of curated variants (in a more explicit form than on p. 11/327ff)?

10/287: can you reason about the sharp drop for the last data point of the duplication model (Fig. S8A)?

10/295: further up on that page, you explain that the top 10 ranking would be a practically relevant performance evaluation, but your statistics now pertain to the top 5 - why?

12/360ff: I am a bit reluctant to accept the word "recent" citing studies from 2019, when the Eichler group in particular has already published an "upgrade" (Science, same as above) to their 2019 Cell paper (Audano et al.). In that context, I am wondering if TADA could also be used in related settings, e.g., help identifying or corroborating SV/QTL associations (cf. the section on "QTL analyses" in the Science paper)?

14/406: I am still a bit fuzzy on what exactly TADA achieves for the use case of a manual CNV prioritization. TADA annotates CNVs with categories and summarizes that info in tabular form (roughly like a series of bedtools operations), is that correct?

14/418ff: at the end of the paper, I have to conclude that the performance of your classification models is solid, and seemingly more robust than competing approaches, but also not stunning. And looking at the feature importances, I am wondering if the TAD-focus of your study adds much. What's the performance of a model that does not use the TAD information?

15/454: the description of the variant sets is hard to follow and seems incomplete. I would like to see a stepwise process, starting from the initial number of variants per data set, followed by the number of variants that were discarded in each filtering step, and then resulting in the final number of variants per analysis. Additionally, it would be informative to see the size distribution of variants at the beginning and at the end of this process (see also my next point).

16/470: for how many out of all variants did you have to lift the coordinates over (and from which assembly)? did you lose variants during the liftover process, and if yes, how many?

17/509ff: if I understand your procedure correctly, you try to create an independent evaluation data set using the ClinVar variants, but your reciprocal overlap criterion is quite strict. How many variants are in the ClinVar data set that show a reciprocal overlap with your training data between, say, 50% and 90% and are thus kept in your evaluation set? Your statement on page 8/220f actually means "no [reciprocal] overlap [of more than 90%]", is that correct? If so, the high ROC-AUC values for the ClinVar data may be overoptimistic because of the remaining variants that still have substantial overlap. What is the model performance when you exclude variants using a less stringent threshold (say, 50% RO)?

18/535: which version and annotation set (basic, comprehensive etc.) of GENCODE did you

use?

21/647: how did you derive these fixed values?

Fig. S2-I: I can't really see any details in this panel. Since this is a supplementary figure, I suggest you use a full page to display the ideogram.

Fig. S4: the short title (?) suggests that this figure is about ChromHMM states, but the HAR annotation is also listed here. Is that a labeling mistake?

List of references: there are numerous references linking to biorxiv instead of the published work (e.g., 14, 16, 25), please carefully revise your list of references

Comments regarding the TADA github repository:

I clicked through the TADA repository, and I was quite delighted to see a well-structured and well-written code base that also includes trivial but often overlooked details such as a software license file. I hope you keep up the good work for the next development cycles of TADA. I have only minor suggestions for improvements that you can address at your own discretion:

- README/setup: the command "python setup.py install" needs to be executed inside the "TADA" folder after cloning, otherwise it leads to a file not found error.
- Running "python setup.py test" leads to a deprecation warning on my system, and the recommendation to run testing with a library such as tox; I can only second that.
- Publishing TADA as a package via bioconda will likely help distribute TADA and grow its user base.
- It would be helpful to state rough resource requirements (CPU hours, RAM, other relevant quantities if applicable) for the analysis in your paper as a guideline for users who want to train their own classifier.

Reviewer reports:

Reviewer #1: The manuscript submitted by Hertzberg and colleagues presents a new method (TADA) for predicting the pathogenicity of Copy Number Variants. The paper address one important problem in the field. I believe that the manuscript could be considered for publication after addressing the comments reported below.

Major Revisions

1) The description of the composition of the datasets use to test TADA should be improved. The authors should include a table summarizing the number of variants in each dataset and their compositions in terms of classes. To make the work reproducible the authors should provide as supplementary materia a file including the all variants included in each dataset with their annotation.

We extended our description of the variant sets used for training and testing in the method section of the revised manuscript. Additionally, we included supplementary figure S16 in which we show the variants count pre- and post filtering for each data set. While we agree with the reviewer that all variants used for training and testing should be made available to the public to make our work reproducible, we are not allowed to share the DECIPHER variants as they are protected by a Data Access Agreement. Still, we provide all other variants (merged common CNVs and filtered ClinVar) in the data subdirectory of our Github repository (<https://github.com/jakob-he/TADA/tree/master/tadacnv/data>).

2) In the datasets used for trining and testing you also considered the "likely pathogenic" variants. In general this annotation is obtained using predictive methods therefore such mutations could be potentially easier to predict. Could you please report the performance of the TADA excluding such mutations from the testing set? Is there any differences in the performance?

We thank the anonymous reviewer for raising this important point. The DECIPHER variants are categorized by the clinicians who uploaded them in five different classes: Pathogenic, Likely Pathogenic, Uncertain Significance, Likely Benign and Benign. These classes generally correspond to posterior probability thresholds referring to their pathogenic potential (shown for example in the pathogenicity tooltip at <https://www.deciphergenomics.org/patient/394902/genotype/174151/browser>). However, the categorization is made by the uploading clinician for each individual variant. Still, we do agree with the reviewer that the class of some variants could be the results of a predictive algorithm and could potentially influence our classification performance. We asses this in a new analysis leaving out 'Likely Pathogenic' DECIPHER variants, repeating the size-matching with common variants, and retraining our model. The changes in ROC-AUC are shown below.

Deletions:

5-CV: 0.8379 -> 0.8222

Test-Split: 0.8059 -> 0.8248

Duplications:

5-CV: 0.8069-> 0.8051

Test-Split: 0.7868-> 0.7960

The results of this analysis show no significant positive influence of 'Likely Pathogenic' variants on our predictive performance. The performance on the test-split of the DECIPHER data even improved slightly. Thus, we argue that our classification is not influenced by the possibility that some variant classes were potentially obtained via predictive methods.

3) In the discussion, it is mentioned that the prediction of the Haploinsufficiency is the most discriminating feature. Can you please report the performance of the method using different combinations of the 14 features to show what is gain in terms of AUC when adding less discriminating features?

We agree with the reviewer that the effect of individual features on our classification is an important part of our analysis. To allow the reader to assess this, we had reported the loss in accuracy of all features after permutation in Figure 4. The combined decrease in accuracy of feature combinations can be estimated based on this data. To better answer the point raised by the reviewer, in the revised manuscript we now also include a graph representation of the partial correlation between features to show which feature combinations provide similar discriminating information (Supplementary Figure S14).

4) As you have pointed out the length of the CNV is a critical feature for the predictions. What is the performance of the method if you split the training and testing sets in short and long CNVs?

To address this point, in the revised manuscript we extend our previous comparison of TADA and SVSCORE with ClinVar variants split by length. We now include SVFX in the comparison and split the ClinVar deletions into four non-overlapping groups: Small (< 50kb), Medium (<100kb), Medium-Large (<1mb), Large (>=1mb). The results of the analysis are shown in Supplementary Figure S6 and described on p.9 l.259-265. TADA outperforms all other methods for Small to Medium-Large sized deletions. There are slight changes in ROC-AUC scores between size groups. However, especially for Large to Medium-Large deletions TADA's performance decreases notably less than the performance of the other two methods, indicating higher robustness towards CNV size as a discriminating feature.

5) A possible solution to mitigate the bias due to the annotation features is to perform a chromosome-based cross-validation procedure. If you keep all the variants of a chromosome in the same subset do you observe any differences in the performance?

The reviewer raises the important point that there might be a difference in density/feature distribution across chromosomes which could influence a classification method. However, in our analysis, we normalize features across the entire training set regardless of genomic position (p.20-21 l.630-633). Any bias introduced due to changes in chromosome-specific feature distribution will be mitigated by this procedure. We argue that it is necessary to

include this normalization not only for scaling differences between features but also to allow for the model to focus on chromosome-independent information rather than a classification.

Minor revisions

1) Many numbers in the text are reported without any digits before the decimals. I suggest to include 0 when needed.

We adjusted the manuscript to reflect the reviewer's suggestion.

2) In the Annotations section at page 20 you mentioned that the selected the P-O-interactions were selected assuming a p-value=3. I believe this is a typo?

We thank the reviewer for spotting this potentially misleading description. The p-value is log-transformed. We now clarify this in the corresponding section of our manuscript (p.20 l.600).

3) Please specify better what does it mean aggregated PhastCons and PhyloP scores? Do you use a combination of them? What are the number of species included in the alignment used for the calculation of such scores?

In the revised manuscript we have included an extended description of our use of base-wise conservation scores (p.19 l.584-589).

Reviewer #2: Hertzberg et al., proposed a new method named TADA for prioritizing putative pathogenic CNVs. In principle, the authors established a random forest model using multiple genomic features including TAD boundaries, FANTOM5 enhancer annotation, genomic conservation, CTCF binding, and haploinsufficiency and intolerance of loss-of-function mutations for genes, etc. To train and test the model, they collected both non-pathogenic and pathogenic CNVs (including deletion) from DECHIPHER, ClinVar CNVs, and GnomAD. The model proposed several important features in classification, mostly associated with the pathogenic score of associated genes. However, such a result is expected due to the biased data collection between coding and non-coding associated pathogenic variants, thus it is hard to conclude that the identified features provide additional insight to annotate pathogenic CNVs. Further, the authors claimed that TADA is accurate than other existing methods and emphasized the utility of the new tool in predicting the disease impact of large genomic alterations, but the currently presented results may not be sufficient to evaluate the validity of TADA.

Major comments

- It is difficult to figure out the uniqueness of TADA compared to other benchmarked algorithms. For example, the authors demonstrated that SVFX is a classifier relying on somatic variants as a proxy for pathogenicity, but the SVFX model also generated a random-forest classifier based on data-driven approaches with various genomic

features similar to TADA. Despite the slight difference in feature sets, it seems SVFX and TADA use essentially the same framework. To deliver the difference more clearly, the authors should evaluate the performance for the same data set used in SVFX and also provide specific examples and statical analysis to demonstrate how each method offers different pathogenic annotations.

We thank the reviewer for raising these points and agree that our previous comparison to SVFX and the description of its limitations could have suggested little difference between the methods. To address this, we have significantly altered the manuscript. In the revised document we present multiple new sections:

1) We extend our description of the practical limitations of SVFX in the Background (p.2-3 l.39-48) as well as the Method (p.21-22 l.660-671) section. We now clarify that SVFX employs a normalization method designed to mitigate the effect of the size difference between pathogenic and non-pathogenic variants. This normalization method is applied to the entire data pre-split into training and test-set, leading to information leakage. The reported ROC-AUC/PR-AUC values by SVFX are therefore overestimating their actual predictive performance. The most striking example is their reported performance on the ClinVar data set (0.99 ROC-AUC/PR-AUC). In our comparison using the ClinVar variants as a test-set, the SVFX model without normalization and size feature results in a ROC-AUC of 0.8311 and 0.7384 for deletions and duplications, respectively.

2) We included a new performance comparison between TADA, SVFX and SVSCORE evaluating their predictive power with regards to the size of CNVs. The results are shown in Supplementary Figure S6. Since SVFX adds size as a feature after the z-score normalization, the likely introduces bias into the model, underestimating the pathogenic effect of smaller to medium-sized variants and overestimating their performance on variants sets with significant size differences between pathogenic and non-pathogenic variants. TADA outperforms SVFX in every size group of ClinVar deletions. Supporting the results of this analysis, the authors of SVFX report size as the most discriminating feature between pathogenic and non-pathogenic variants.

The previous two points lead to our initial decision to compare SVFX and TADA based on our size-matched dataset without size as a feature since their reported method to mitigate size bias cannot be applied and with size-matched data the size feature should not be necessary. The data for the germline CVD and IBD cohort used in their analysis is, in contrast, not size-matched and would require normalization.

In conclusion, we agree with the reviewer that method structure (i.e. data-driven random forest classifiers and feature composition) of TADA and SVFX are similar but SVFX shows practical limitations especially with focus on the size bias between pathogenic non-pathogenic variants while, based on our new analysis, TADA performs well for variants of all sizes.

- CNVs are very common in the genome, even for healthy individuals. What happens if the authors provide only non-pathogenic germline CNVs as an input? Does TADA recognize all these germline variants as non-pathogenic?

In our manuscript, we focus on TADAs performance on both classes of variants since we use primarily AUC-ROC values as performance measurements. However, in our comparison

with VEP we provide F1-Scores averaged over the individually measured predictive performance on non-pathogenic and pathogenic variants. To address the point raised by the reviewer, we now include these individual scores in our method section for the Test-Split CNVs (p.21 l. 653-659) and list the corresponding F1-Scores below. The F1-scores indicate a similarly high predictive performance of TADA for pathogenic and non-pathogenic variants.

Deletions:

Test-set performance:

Non-Pathogenic:	0.73
Pathogenic:	0.75

Duplications:

Test-set performance:

Non-Pathogenic:	0.71
Pathogenic:	0.75

- As pointed by the authors, the method is highly biased to coding sequence associated variants due to the biased data collection. However, the mechanism of action of coding-sequence and noncoding-sequence associated variants should be different. Thus, the authors should test how the performance differs when using only coding sequence associated variants and non-coding sequence associated variants.

We agree with the premise of the reviewer's comment in that the mechanism of action of coding versus noncoding sequence-affecting variants should be different. Unfortunately, as we also discuss in our manuscript the current publicly available catalogue of pathogenic CNVs consists almost exclusively of variants affecting the coding sequence. Given this lack of publicly available data, it is currently not possible to construct a set of non-coding sequence associated variants to test our performance. A simulation of such a non-coding sequence associated variant set would require an extensive amount of work and is out of the scope of this methodological manuscript. Still, we expect that with newly assayed non-coding annotation as well as pathogenic non-coding variants the role of the features describing the CNV's regulatory impact aside from coding regions in our classifier will increase manyfold.

- To make the prediction result more interpretable, I'd like to suggest include several example genome browsers snapshots and demonstrate how the variants are recognized as pathogenic and what is the expected mechanism.

We now include a new analysis of two developmental disease (DD) patients with duplications found to be pathogenic in a previous publication by Melo et al. 2020 (p.11-12 l.335-358). TADA performs well, predicting both duplications as pathogenic and ranking them higher than the 90th percentile of pathogenicity scores among all other duplications identified in individual DD2. To address the point raised by the reviewer, we further explore the regulatory environment of the two disease-causing duplicates using IGV. Guided by the information in the IGV screenshot (Supplementary Figure S13) we speculate about TADAs potential decision leading up to the predicted pathogenicity score.

- The choice of feature data set could be important in classification performance. For example, TAD annotation is entirely arbitrary. Depending on the hard cutoff or parameters used, the TAD annotation could also identify sub-TAD structures. As exemplified in TAD annotation several genomic features are highly variable depending on the user-defined parameters, the authors should show how such variations affect the robustness of TADA.

As the reviewer correctly points out, TAD boundary annotation is highly variable across calling methods as well as parameter setting of the individual approaches. We show in our manuscript that with our current deletion model the importance of TAD boundaries is the least discriminative feature between pathogenic and non-pathogenic variants. In its current state the TAD environment and its annotation serves - as described in the TADA-Workflow - primarily as a computational measurement to improve run time that also approximates the regulatory environment. Any changes to the TAD definitions will therefore only result in minor changes to our predictive performance. To test this assumption, we now conduct an additional analysis replacing the ES TAD boundaries with "chromosome-wide" TAD i.e. using TADs spanning the entire chromosome. The resulting drop in ROC-AUC compared to the classifier trained on ES TADs is 0.0110 and 0.0160 for the deletion and the duplication model, respectively. This analysis shows, as expected, that with current TADA framework changes to the TAD definition will not influence the predictive performance of our models.

- Page 13 line 391 'significant depletion of pathogenic variants in TAD boundaries~' : such interpretation should be biased by the over-collected coding-sequence-associated pathogenic variants.

We entirely agree with the reviewer in that the significant depletion of pathogenic variants in TAD boundaries is likely biased by the over-collected coding-sequence-associated pathogenic variants and address this issue in our manuscript on p.14-15 l.432-445.

- As shown in Figure 4, all essential features are associated with pathogenic impact on involved genes. This result might imply that the essential part of TADA classification is known pathogenicity of the corresponding gene disruption rather than other genomic features. Suppose these genomic features such as TAD boundary, CTCF distance, enhancer distance are not critical in TADA. In that case, it is unclear the advantage of TADA in classifying pathogenic variants rather than simple classification based on the pathogenic effect of affected genes

We thank the reviewer for the insightful comment and agree that the relative role of non coding versus coding features in current pathogenicity algorithms is not assessed in depth. Most currently available training sets consist of pathogenic coding variants, which means the current Random Forest classifiers trained on the DECIPHER data do rely to a large degree on gene centric features as shown and discussed in Figure 4. The most important features focus on individual genes (Gene HI) and the combination of affected genes (HI-Log-Odds Score), suggesting that the model does classify genes in order to classify CNVs but also incorporate the information of multiple coding regions rather than just the most severe score out of all genes. Since the current training data almost entirely consist of pathogenic coding variants, as indicated in the enrichment analysis, the benefit of non-coding annotation for the classification is marginal. However, we are not aware of any methods similar to TADA or the

approaches included in our performance comparison that reliably prioritize pathogenic CNVs or SVs in general, even if only based gene centric information. Given this lack of methods to annotate and classify CNVs based on their pathogenic affect, we argue that TADA serves as a meaningful contribution to the study of genomic alterations and potentially in clinical diagnostics and will only increase in importance once large non-coding data becomes accessible to the community.

Reviewer #3: Hertzberg et al. present the new tool "TADA" for the manual or machine learning-assisted prioritization of unbalanced SVs (CNVs; deletions and duplications) in terms of their pathogenic potential. The authors perform an extensive enrichment analysis to identify genomic annotation categories (genes, enhancers etc., and restricted by the local regulatory environment defined by TAD boundaries) that are associated with CNV pathogenicity. The identified annotations are then used to inform feature design for the machine learning part of TADA. The classification models are trained and evaluated on different sets of curated CNVs and seem to deliver a more stable and often better performance than competing approaches. The authors acknowledge that there are several factors that likely impact the general applicability of TADA (e.g., detection and annotation biases for the curated CNVs) and that are hard to quantify. Overall, TADA is a thoroughly designed and tested tool that tackles an important problem. Since a main output of the (automated) TADA analysis is a ranked list of potentially pathogenic CNVs, its output needs further evaluation by a domain expert.

Below is a list of more specific comments:

Comments:

2/41: I am unclear about your criticism of the SVFX tool. Is that speculation, or do you substantiate that claim somewhere?

We thank the reviewer for pointing out this lack of clarity. We have now substantially altered the relevant sections (p.2-3 l.39-48; p.21-22 l.660-671). Also, we provide additional evidence for our claims in our size-dependent performance evaluation in Supplementary figure S6.

3/56 "biological relevance": your use of the expression "biological relevance" seems a bit overly assertive in the given context. The enrichment tests suggest an association, but the actual relevance can commonly only be assessed in the wet lab.

We thank the reviewer for pointing out this potentially misleading phrase. We adjusted the manuscript accordingly (p.3 l.61).

4/85: any specific reason why you did not use the large set of SVs published a couple of months ago in Science ("Haplotype-resolved diverse human genomes and integrated analysis of structural variation") when compiling your data set?

Since we submitted the first version of our manuscript to an alternative journal in 2020, our initial analysis was performed before Audano et al published their updated set of SVs. However, we agree with the reviewer that this set of variants could potentially aid in the

predictive performance of our model. We therefore set out to investigate the changes to our set of common CNVs if including the new catalogue of CNVs obtained from Audano et al. 2021. Starting from a total of 111,480 variants, we filtered for deletions with an allele frequency greater than 0.1, located on autosomes. We overlapped the remaining variants to our current set of non-pathogenic CNVs, resulting in a set of 14,378 deletions. Critically, the number of variants was reduced to 44 after size-matching with DECIPHER deletions. Thus, we decided to refrain from updating our analysis results. As we show in Supplementary Figure 1 and 2, this drastic decrease in variants is due to the difference in the size distribution of pathogenic and non-pathogenic variants. The majority of size-matched variants consist of BioBank and DGV variants. We expect that with annotated pathogenic variants from WGS - especially long-read WGS - the number of size-matched variants from GnomAD-SV, Audano et al. (both 2019 and 2021) will only increase leading to a more robust basis for our pathogenicity predictions.

4/106: please make explicit what quantity the q-value is designating (re: multiple testing/FDR).

We adjusted the manuscript accordingly (p.4 l.104-107).

5/135: please be more verbose on why you think the observed depletion at TAD boundaries is "interesting".

We again thank the reviewer for highlighting this imprecise statement. We have now altered the manuscript to clarify this. In the revised version, we replaced the specific sections pointing out that CNVs have been found to cause disease phenotypes by disrupting TAD boundaries, which is in contrast to the observed significant depletion (p.6 l.142-143). This is likely due to the focus on gene sequence affecting variants in our data set as discussed further in the results and discussion sections.

6/140ff: I have trouble understanding why you have that break before adding HAR, ChromHMM and SD annotations to your enrichment analysis? None of those categories seem far-fetched when looking for potential associations to CNVs. Also, it is hard to imagine that any set of current genomic annotations would verifiably represent the full (!) spectrum of regulatory activity. Maybe you make a distinction here that I have missed?

In the initial manuscript we decided on this break to emphasize the first set of annotations since only they were further included in our classification. However, we fully agree with the reviewer that HAR, ChromHMM and SD annotation are reasonable candidates to describe the regulatory environment affected by CNVs and in this section of the manuscript they are equally important compared to the first set of annotations. So in the revised manuscript we removed the contextual break (p.6 l.149-151).

7/192ff: indeed; can you summarize what potential (annotation) biases could be in your set of curated variants (in a more explicit form than on p. 11/327ff)?

We now include a more extensive description of the biases in our set of curated variants causing the increased focus on coding-sequence features (p.12 l.363-369).

10/287: can you reason about the sharp drop for the last data point of the duplication model (Fig. S8A)?

As described in the Methods section the number of duplications is considerably smaller than the number of deletions (p.17 l.528-532). The sharp drop for the last data point is likely due to an unevenly distributed set of duplications in the test set in terms of pathogenicity score. Only a small proportion of duplications in this specific test set received a very high pathogenicity score. In contrast, the number of deletions in the test set is high enough to reliably represent the entire spectrum of pathogenicity scores.

10/295: further up on that page, you explain that the top 10 ranking would be a practically relevant performance evaluation, but your statistics now pertain to the top 5 - why?

We apologize for this potentially confusing description and thank the reviewer for spotting this. In a previous analysis we used the first ten ranks rather than the first five to compute the corresponding statistics. We have now altered this, and correctly refer to the first five ranks in our results section.

12/360ff: I am a bit reluctant to accept the word "recent" citing studies from 2019, when the Eichler group in particular has already published an "upgrade" (Science, same as above) to their 2019 Cell paper (Audano et al.). In that context, I am wondering if TADA could also be used in related settings, e.g., help identifying or corroborating SV/QTL associations (cf. the section on "QTL analyses" in the Science paper)?

We entirely agree with the reviewers' reluctance to describe the studies as "recent". The phrase in our revised manuscript accordingly (p.13 l.399-400). The eQTL Analysis of Audano et al. is focused on finding SV-eQTL for common diseases. While it would be possible for TADA to classify these variants as pathogenic or non-pathogenic, it is unlikely that they will be annotated as highly pathogenic since the DECIPHER data set is focused on more pronounced developmental phenotypes.

14/406: I am still a bit fuzzy on what exactly TADA achieves for the use case of a manual CNV prioritization. TADA annotates CNVs with categories and summarizes that info in tabular form (roughly like a series of bedtools operations), is that correct?

The use of TADA's manual CNV prioritization highly depends on which set of annotations the user specifies. If the default annotation set is used, TADA computes the 14 features also used for our predictive algorithm. Based on these features the user can manually sort the variants, for example, based on the HI-score of affected genes and decide on appropriate thresholds. In this use-case the manual prioritization, therefore, extends beyond overlap operations. However, if the user provides individual sets of genomic elements as annotation, the output is limited to CNVs annotated with distances for each set of genomic elements, which can be compared to performing a series of bedtools operations.

14/418ff: at the end of the paper, I have to conclude that the performance of your classification models is solid, and seemingly more robust than competing approaches, but also not stunning. And looking at the feature importances, I am wondering if the TAD-focus of your study adds much. What's the performance of a model that does not use the TAD information?

In the manuscript we show that with our current deletion model the importance of TAD boundaries is the least discriminative feature between pathogenic and non-pathogenic variants. In its current state the TAD environment and its annotation serves - as described in the TADA-Workflow - primarily as a computational measurement to improve run time that also approximates the regulatory environment to an extent. To test this assumption, we now perform an additional analysis replacing the ES TAD boundaries with "chromosome-wide" TADs i.e. using TADs spanning the entire chromosome. The resulting drop in ROC-AUC compared to the classifier trained on ES TADs is 0.0110 and 0.0160 for the deletion and the duplication model, respectively. This analysis shows, as expected, that with current TADA framework changes to the TAD definition will not influence the predictive performance of our models. However, we argue, as with all types of non-coding annotation, the influence of TAD definitions on the prediction of pathogenic variants is limited by our current training data and will increase with a higher proportion of pathogenic non-coding-sequence affecting variants.

15/454: the description of the variant sets is hard to follow and seems incomplete. I would like to see a stepwise process, starting from the initial number of variants per data set, followed by the number of variants that were discarded in each filtering step, and then resulting in the final number of variants per analysis. Additionally, it would be informative to see the size distribution of variants at the beginning and at the end of this process (see also my next point).

16/470: for how many out of all variants did you have to lift the coordinates over (and from which assembly)? did you lose variants during the liftover process, and if yes, how many?

To address both of the above mentioned points by the reviewer, we now extended our description of the variant sets used for training and testing in the method section of the revised manuscript. Additionally, we included supplementary figure S16 showing the variants count pre- and post filtering for each data set. In Supplementary Figure S17 E we also show the size distributions of each variant set after applying the filters.

17/509ff: if I understand your procedure correctly, you try to create an independent evaluation data set using the ClinVar variants, but your reciprocal overlap criterion is quite strict. How many variants are in the ClinVar data set that show a reciprocal overlap with your training data between, say, 50% and 90% and are thus kept in your evaluation set? Your statement on page 8/220f actually means "no [reciprocal] overlap [of more than 90%]", is that correct? If so, the high ROC-AUC values for the ClinVar data may be overoptimistic because of the remaining variants that still have substantial overlap. What is the model performance when you exclude variants using a less stringent threshold (say, 50% RO)?

We thank the reviewer for this insightful comment and for pointing out a potential methodical issue in our manuscript. To assess if the predictive performance of our model is influenced by the reciprocal overlap threshold when comparing our training data to the ClinVar variants, we now include an additional analysis. The results are shown in Supplementary Figure S20 and described in our methods section (p.18-19 l.566-571). The ROC-AUC values remain similar across the spectrum of reciprocal overlap thresholds (0.1-0.9), indicating no significant influence of ClinVar variants overlapping with our training data on our performance.

18/535: which version and annotation set (basic, comprehensive etc.) of GENCODE did you use?

We added the GENCODE version used for our analysis in the method section (p.19. l.591).

21/647: how did you derive these fixed values?

In the revised manuscript we now elaborate on the reasoning for setting these fixed values as missing value replacements (p.23 l.708-712).

Fig. S2-l: I can't really see any details in this panel. Since this is a supplementary figure, I suggest you use a full page to display the ideogram.

We thank the reviewer for pointing us towards this issue and, as suggested, now introduce a full page display for the ideogram in our revised manuscript.

Fig. S4: the short title (?) suggests that this figure is about ChromHMM states, but the HAR annotation is also listed here. Is that a labeling mistake?

We corrected the figure caption accordingly.

List of references: there are numerous references linking to biorxiv instead of the published work (e.g., 14, 16, 25), please carefully revise your list of references

We revised the list of references in the revised manuscript and updated entries where necessary.

Comments regarding the TADA github repository:

I clicked through the TADA repository, and I was quite delighted to see a well-structured and well-written code base that also includes trivial but often overlooked details such as a software license file. I hope you keep up the good work for the next development cycles of TADA. I have only minor suggestions for improvements that you can address at your own discretion:

- README/setup: the command "python setup.py install" needs to be executed inside the "TADA" folder after cloning, otherwise it leads to a file not found error.**
- Running "python setup.py test" leads to a deprecation warning on my system, and the recommendation to run testing with a library such as tox; I can only second that.**
- Publishing TADA as a package via bioconda will likely help distribute TADA and grow its user base.**

- It would be helpful to state rough resource requirements (CPU hours, RAM, other relevant quantities if applicable) for the analysis in your paper as a guideline for users who want to train their own classifier.

We appreciate the kind comments by the reviewer. We have now updated the TADA Github repository as follows: We published TADA as a Pypi Package and are currently working on a Bioconda release with the same content. This updated TADA version allows us to predict and annotate variants with our default features and pre-trained models without the use of config files i.e. the user is able to instantly apply our method after pip installation. We also updated the README as suggested and included changes regarding the new TADA release. Since predicting, annotating and even training a new classifier takes rarely more than a few minutes on a standard workstation, there are no minimum resource requirements for TADA that we could include.

Second round of review

Reviewer 1

In the revised version of the manuscript the authors significantly improved the quality of their manuscript. I suggested the Editor to accept the manuscript after addressing the minor points reported below.

Minor Revisions

- 1) What are the gray bars in Fig S1-S5? Please specify.
- 2) In page 9 $\leq 1\text{mb}$ should be read $>1\text{mb}$
- 3) In page 10 when defining calibration (cit 39), I suggest to include a recent paper (Benevenuta et al. 2021 Bioinformatics PMID:33492342)
- 4) In the manuscript is reported that CADD score is aggregated in one of the alternative CNV scoring methods. The authors of CADD recently developed CADD-SV (<https://cadd-sv.bihealth.org/>), can you please include the performance of CADD-SV on your datasets?

Reviewer 2

Hertzberg et al., conducted further in-depth data analysis to address the previously raised concerns and clarify what TADA exactly does. My additional comments are listed below.

1. The authors responded that the current TADA focused on coding sequence pathogenic variants due to the inability to access noncoding associated pathogenic variants in establishing TADA model. If TADA is limited to predict noncoding pathogenic variants and unable to validate its performance, TADA should focus on prediction of coding sequence pathogenic variants, which should be clearly described throughout the entire manuscript.
2. If TADA focused on coding-sequence variants and TAD definition does not affect the predictive performance, the advantages of TADA compared to previously reported methods are not clear.
3. The performance test for both non-pathogenic and pathogenic is similar in terms of F1-scores. Considering that most variants in the genome are non-pathogenic, the current performance of TADA appears to be insufficient to discriminate pathogenic variants from nonpathogenic variants which account for the majority of test variants.
4. The presented genome browser snapshot does not illustrate how TADA predicted the example variant as pathogenic.

Reviewer 3

Hertzberg et al. have substantially revised their manuscript and added the necessary clarifications and analyses to notably improve their work. The authors have addressed all my concerns in a satisfactory manner and I have thus no further questions and comments.

Reviewer reports:

Reviewer #1: In the revised version of the manuscript the authors significantly improved the quality of their manuscript. I suggested the Editor to accept the manuscript after addressing the minor points reported below.

Minor Revisions

1) What are the gray bars in Fig S1-S5? Please specify.

We have clarified the captions of the corresponding figures in the manuscript.

2) In page 9 ζ 1mb should be read >1 mb

We adjusted the manuscript to reflect the reviewer's suggestions.

3) In page 10 when defining calibration (cit 39), I suggest to include a recent paper (Benevenuto et al. 2021 Bioinformatics PMID:33492342)

We thank the reviewer for the helpful suggestion and now include the recommended paper as a reference.

4) In the manuscript is reported that CADD score is aggregated in one of the alternative CNV scoring methods. The authors of CADD recently developed CADD-SV (<https://cadd-sv.bihealth.org/>), can you please include the performance of CADD-SV on your datasets?

We conducted an additional performance comparison against the recently developed method CADD-SV. The results are shown in Supplementary Figure S7 and described in the result section (p.9 l.266-271) as well as in the discussion (p.16 l.470-474). TADA outperforms CADD-SV with regard to both the Test-Set of DECIPHER CNVs and ClinVar variants.

Reviewer #2: Hertzberg et al., conducted further in-depth data analysis to address the previously raised concerns and clarify what TADA exactly does. My additional comments are listed below.

3. The performance test for both non-pathogenic and pathogenic is similar in terms of F1-scores. Considering that most variants in the genome are non-pathogenic, the current performance of TADA appears to be insufficient to discriminate pathogenic variants from nonpathogenic variants which account for the majority of test variants.

The reviewer expresses concern about TADA's performance on non-pathogenic variants measured by F1-scores - especially given their higher abundance in the genome compared to pathogenic variants. Indeed, most variants identified in a patient are likely non-pathogenic and should be identified as such. However, we argue that the F1-Score as a performance predictor for variants-scoring methods does not fully capture the practical problem. TADA and other variant-scoring methods, in general, are designed to assist in the identification of

pathogenic variants during diagnostics. The challenge we aim to address is the identification of a single pathogenic variant among a large background of non-pathogenic variants. The pathogenicity score produced by TADA should be used to guide which variants are to be further analyzed as potential disease-causing candidates, thus reducing the total number of variants investigated in downstream analysis to a manageable number. An ideal classifier would therefore focus on producing a well-calibrated pathogenicity score and rank the true pathogenic variant as high as possible rather than classifying all variants either as pathogenic or non-pathogenic based on an arbitrary threshold. This ranking ability is not measured by the F1-Score. The F1-Score can be interpreted as a harmonic mean of precision and recall and requires binary predictions for each sample. Thus, it is a measure of performance based on a single threshold on a model's prediction probabilities. While it provides an approximation of the model's ability to separate pathogenic from non-pathogenic variants, the calibration of the class probabilities is not considered and the threshold itself needs to be optimized.

In contrast, ROC curves and the corresponding AUC values represent the model's performance for multiple thresholds which allows for a more accurate assessment of the predictive power. However, as described by Benvenuto et al. [1] ROC curves alone are not sufficient to evaluate a variant-scoring method.

In our manuscript, we use ROC-AUC values as the first indicator of predictive power and, to directly simulate the prioritization of pathogenic variants in a clinical setting we then conduct a ranking analysis. We place one pathogenic variant among 99 non-pathogenic variants and assess the model's ability to place the pathogenic variant among the top 5 based on predicted pathogenicity. Rather than any metric involving individual pathogenicity score thresholds, this analysis represents the main indicator of TADA's performance since it approximates its actual mode of operation. The ROC-AUC values and more importantly the ranking analysis presented in our manuscript, clearly demonstrate that TADA is able to reliably distinguish pathogenic from non-pathogenic variants. We have now clarified the corresponding sections in our manuscript (p.10 l.293-305, p.16 l.477-479)

2. If TADA focused on coding-sequence variants and TAD definition does not affect the predictive performance, the advantages of TADA compared to previously reported methods are not clear.

We have in many ways shown "the advantages of TADA": The presented ROC-AUC and ranking performance indicate a clear advantage of TADA compared to other currently available approaches. The performance advantage holds regardless of the reliance on coding information or on the relative contribution of TADs to the prioritization. Secondly, compared to SVScore and VEP, TADA also employs an entirely new framework that allows for annotation of CNVs to assist manual analysis and training of prioritization models with user-defined annotations. While SVFX employs a similar framework to ours it has significant practical limitations, which we have discussed at length in the last revision. Hence, to address the second comment by the reviewer, we argue that the superior performance, as well as the framework capability of TADA, are the main advantages over comparable approaches.

1. The authors responded that the current TADA focused on coding sequence pathogenic variants due to the inability to access noncoding associated pathogenic variants

in establishing TADA model. If TADA is limited to predict noncoding pathogenic variants and unable to validate its performance, TADA should focus on prediction of coding sequence pathogenic variants, which should be clearly described throughout the entire manuscript.

We agree with the reviewer that the reliance of TADA on coding information can be considered a limitation. As described in the manuscript and the previous revision, there are no non-coding pathogenic variants publicly available to train or validate a machine learning model. However, if we built a classifier based solely on coding information we would be blind to the known importance of non-coding features when in the future such variants are identified [2]. The inclusion of non-coding features in the manuscript helps us to gain mechanistic insight - as presented in the enrichment analysis - and sensitizes us to the importance of non-coding pathogenic variants. We designed the TADA framework to be able to retrain our model as soon as pathogenic non-coding CNVs are publicly available. Additionally, we allow for manual analysis of potentially pathogenic variants based on non-coding features. Yet, we are well aware that we cannot currently validate the non-coding features via an individual call set. Following the reviewer's comments, we have now amended our manuscript to better emphasize this limitation throughout our manuscript (p.11 l.336-340, p.13 l.386-396, p.14-15 l.435-440, p.15 l.442-455, p.15-16 l.466-470, p.16 486-496). In our discussion section we now explicitly recommend the application of TADA with a focus on coding rather than non-coding information.

4. The presented genome browser snapshot does not illustrate how TADA predicted the example variant as pathogenic

The aim of including the genome browser screenshot of the two developmental-disease (DD) associated duplications, that serve as examples of TADA's prediction process, is to visualize their regulatory environment given the annotations underlying our feature set. As further discussed in the final paragraph of our result section (p.12-13 l.369-396) we approximate the contribution i.e. loss of accuracy of each feature to the prediction process by permutation. Based on the results of this analysis we are able to speculate about the features driving the decision process for both DD-associated duplications. As a consequence of the reviewer's helpful comment, we have now adjusted the discussion of the two duplications in our manuscript, clearly referencing the feature importance and the corresponding figure (p.12 l.341-367).

[1] Benevenuta, Silvia, Emidio Capriotti, and Piero Fariselli. "Calibrating variant-scoring methods for clinical decision making." *Bioinformatics* 36.24 (2020): 5709-5711.

[2] Spielmann, Malte, and Stefan Mundlos. "Looking beyond the genes: the role of non-coding variants in human disease." *Human molecular genetics* 25.R2 (2016): R157-R165.